# Hard Ticks as Vectors: The Emerging Threat of Tick-Borne Diseases in India

**DOI:** 10.3390/pathogens13070556

**Published:** 2024-07-02

**Authors:** Nandhini Perumalsamy, Rohit Sharma, Muthukumaravel Subramanian, Shriram Ananganallur Nagarajan

**Affiliations:** Division of Vector Biology and Control, Indian Council of Medical Research—Vector Control Research Centre (ICMR-VCRC), Puducherry 605006, India; pnandhini015@gmail.com (N.P.); rohitpuducherry@gmail.com (R.S.); kumaravelmuthuvel@gmail.com (M.S.)

**Keywords:** hard ticks, distribution, life cycle, salivary glands, vector competence, tick-borne pathogens, climatic change

## Abstract

Hard ticks (Ixodidae) play a critical role in transmitting various tick-borne diseases (TBDs), posing significant global threats to human and animal health. Climatic factors influence the abundance, diversity, and vectorial capacity of tick vectors. It is imperative to have a comprehensive understanding of hard ticks, pathogens, eco-epidemiology, and the impact of climatic changes on the transmission dynamics of TBDs. The distribution and life cycle patterns of hard ticks are influenced by diverse ecological factors that, in turn, can be impacted by changes in climate, leading to the expansion of the tick vector’s range and geographical distribution. Vector competence, a pivotal aspect of vectorial capacity, involves the tick’s ability to acquire, maintain, and transmit pathogens. Hard ticks, by efficiently feeding on diverse hosts and manipulating their immunity through their saliva, emerge as competent vectors for various pathogens, such as viruses, parasites and bacteria. This ability significantly influences the success of pathogen transmission. Further exploration of genetic diversity, population structure, and hybrid tick vectors is crucial, as they play a substantial role in influencing vector competence and complicating the dynamics of TBDs. This comprehensive review deals with important TBDs in India and delves into a profound understanding of hard ticks as vectors, their biology, and the factors influencing their vector competence. Given that TBDs continue to pose a substantial threat to global health, the review emphasizes the urgency of investigating tick control strategies and advancing vaccine development. Special attention is given to the pivotal role of population genetics in comprehending the genetic diversity of tick populations and providing essential insights into their adaptability to environmental changes.

## 1. Introduction 

Arthropoda stands out as the largest and most diverse phylum within the animal kingdom, encompassing crustaceans, arachnids, and insects. Insects, a remarkably varied group, play vital roles in ecosystems by contributing to pollination and honey production, and serving as a food source [1,2]. However, hematophagous insects like mosquitoes, sandflies, and triatomine bugs take on a different role, transmitting pathogens and acting as vectors for several human diseases such as malaria, dengue, zika, chikungunya, leishmaniasis, sleeping sickness, and Chagas disease [3,4]. Within blood-sucking arthropods, “Ticks” hold the distinction of being the second most significant vectors of human diseases [5,6,7,8]. Ticks, belonging to the class Arachnida, which also includes spiders, scorpions, and mites, [9,10,11,12] differ from insects with their non-segmented and spherical bodies [13,14]. The two main types of ticks are hard ticks (Ixodidae) and soft ticks (Argasidae), with an additional monotypic family known as Nutalliellidae [15,16,17].

Hard ticks, as obligate blood feeders, stand out as primary vectors capable of transmitting a broader array of pathogens than any other arthropod vector globally [18,19]. Infesting various vertebrates, they play a pivotal role in transmitting infections to both humans and animals, leading to a spectrum of tick-borne diseases (TBDs) [6,17,20]. Ixodid ticks, during a single feeding, can consume substantial amounts of host blood, thereby increasing the likelihood of acquiring pathogens from infected hosts [8,21,22]. Notably, during blood feeding, ticks inject a significant amount of saliva containing immunomodulatory proteins to impair the host’s anti-tick response [23,24]. Factors such as a long lifespan, high reproductive capacity, a broad host range, and the ability to survive in harsh conditions contribute to making ticks efficient vectors for various infections [6,25,26,27]. Despite extensive research on mosquitoes, mosquito-borne diseases, and mosquito control, there is a noticeable dearth of studies focused on ticks, underscoring their equal significance as disease vectors [28,29].

Tick vectors, with ubiquitous distribution, are expanding globally, posing significant public health risks linked to emerging TBDs [30,31,32]. Notable hard ticks that commonly bite humans include *Ixodes scapularis*, *Amblyomma americanum*, *Dermacentor variabilis*, *Ixodes pacificus*, *Amblyomma hebraeum*, *Hyalomma anatolicum*, *Hyalomma marginatum*, *Haemaphysalis spinigera*, *Ixodes ricinus*, *Ixodes persulcatus*, *Ixodes holocyclus*, and *Dermacentor andersoni*, while soft ticks comprise *Otobius megnini* and *Ornithodoros coriaceus* [33,34,35]. These tick species are commonly encountered in areas where people come into contact with ticks [35]. They are crucial as vectors for certain diseases.

There are records of at least five hard tick species (*Amblyomma integrum*, *Haemaphysalis spinigera*, *Dermacentor auratus*, *Rhipicephalus haemaphysaloides*, and *Hyalomma isaaci*) and one soft tick species (*Otobius megnini*) infesting people in India [36,37]. Researchers have sporadically observed zooanthrophilic *Hyalomma* species, such as *H. marginatum* and *H. truncatum*, feeding on humans [38]. Typically, these ticks target individuals exposed to vegetation and wooded areas, underscoring the importance of personal protection, routine tick checks, and prompt removal for effective prevention of TBDs [29].

In India, notable TBDs encompass Kyasanur Forest Disease (KFD), Crimean–Congo hemorrhagic fever (CCHF), Lyme disease (LD), Q fever (coxiellosis), and Rickettsial infections [39,40,41,42]. The first reported outbreaks of KFD occurred in Karnataka in 1957, with subsequent positive sero-surveillance findings documented in various states across India, despite the absence of major outbreaks [39,43,44,45,46]. Concerns regarding the prevalence of CCHF exist in Gujarat, Rajasthan, and Uttar Pradesh, with positive cases identified in both human and animal samples across the country [47,48,49,50]. Given these circumstances, a critical focus on TBDs is essential, necessitating the implementation of effective tick control measures [51]. Ongoing studies predominantly center on tick control methods, incorporating chemical, biological, and immunological approaches, owing to the transmission of various diseases to both humans and animals [26,52,53]. 

The occurrence and outbreak of TBDs exhibit variability depending on climatic conditions, including the presence of virulent pathogens, abundant competent vectors, diverse animal hosts, suitable environmental settings, and vulnerable human populations [25,54,55,56,57]. Given that tick populations are highly localized, factors such as global warming contributing to climate change may escalate the threat of TBDs, heightening the potential for outbreaks [16,58]. The recent outbreaks of severe viral infections, notably CCHF in India [49,50], are concerning, especially due to the presence of a diverse range of hard ticks serving as competent vectors. Insufficient research exists on competent vectors, their geographical distribution, and the potential expansion of known vectors in India. The growing apprehension about tick-borne infections in the country, coupled with the expanding host range of tick vectors, has prompted an analysis of available data on TBDs. This review article delves into the distribution and diversity of hard tick vectors, exploring their biology, expansion patterns, vector competence, and control strategies. Additionally, the review highlights the epidemiology, transmission dynamics, and potential future risks associated with TBDs in India.

### 1.1. Tick Vectors: An Introduction 

Ticks, originating 225 million years ago as parasites of reptiles during the pre-mid cretaceous period, have evolved from their ancient origins [6,59,60]. Taxonomically, ticks are closely related to mites and fall within the Arachnida class, which also includes spiders and scorpions. Three families classify them: Argasidae (soft ticks), Ixodidae (hard ticks), and the monotypic Nutalliellidae [61,62,63] (Figure 1). There are approximately 109 tick species distributed across 12 genera in India [33,64,65].

The Ixodidae family, or hard ticks, is the dominant family in terms of species diversity and medical and veterinary significance. With over 650 species organized into two major groups (Prostriata and Metastriata), five subfamilies, and 13 genera, hard ticks play a crucial role in disease transmission [67,68,69,70]. As a result of taxonomic re-evaluation, the family Argasidae, ranked second to Ixodidae, has been divided into two subfamilies, Ornithodorinae and Argasinae [71,72,73]. Argasidae, or soft ticks, consist of around 170 species, predominantly found in sheltered environments such as caves, animal nests, burrows, and human habitats. They are characterized by their leathery integument and lack of a dorsal shield (scutum) [73,74]. Some of these ticks serve as vectors for relapsing fevers caused by *Borrelia* spp. in humans and can transmit arboviruses and other bacterial pathogens [75,76]. The third tick family, *Nuttalliella namaqua* (only species of Nuttaliellidae), is a unique species bridging the characteristics of hard and soft ticks [62]. *N. namaqua*, considered the “evolutionary missing link” between the two tick families, was previously classified within the hard tick family, specifically related to the genus *Ixodes* [62,63,77,78]. Hard ticks serve as the main carriers of viruses and bacteria significant to the medical and veterinary field [30,79].

### 1.2. Structure and Physiology of Hard Ticks 

Ixodidae, commonly known as hard ticks, are categorized into two groups, Prostriata (e.g., *Ixodes* spp.) and Metastriata (e.g., *Hyalomma excavatum* and *Rhipicephalus sanguineus*), based on morphological distinctions [67,80,81]. Prostriate ticks are identified by a prominent anterior anal groove, while metastriate ticks feature a small posterior slit-like anal groove [70,82].

The life cycle of Ixodid ticks comprises four stages: egg, larva, nymph, and adult [70]. Females possess a rigid cuticular plate, or scutum, on the front part of the dorsal body surface, whereas in males, the scutum covers nearly the entire dorsal surface [70,82]. Metastriata hard ticks have ornate patterns on their scutum and festoons that help identify them at the species level, but it is still not clear what their biological purpose is [83]. Ticks have a distinct body structure characterized by the anterior capitulum, housing their mouthparts, and the posterior idiosoma, where the legs attach, along with the genital pore, spiracles, and anal aperture [84] (Figure 2). In the adult stage, key features include the genital aperture situated between coxa III, spiracles positioned adjacent to coxa IV, and the presence or absence of eyes in immature and adult ticks. Larvae have six legs, while nymphs and adults each have eight legs [70,82]. Adult ticks have well-developed pulvilli, four separate plates (ventral, anal, and two lateral), and possibly festoons with a scalloped pattern along the back edge. Larvae show an anterior scutal shield, and the entire dorsal capitulum is visible from above [5,85]. The historical understanding of tick biology dates back to texts like Pliny the Elder’s “Historia Naturalis” and Dr. Thomas Moufet’s “Insectorum sive Minimorum Animalium Theatrum”, which noted the absence of a waste elimination mechanism in engorged ticks [86]. The synganglion is the tick’s central nervous system. It is located anteroventrally above the genital pore and controls many bodily functions. It is the fused central nervous system that controls organ activity and coordinates responses [87,88]. Respiration in ticks occurs through numerous tiny air tubes connected to paired spiracles, enabling essential gas exchange for their survival [89]. A simple heart, mid-dorsally located, filters and circulates hemolymph, a vital circulating fluid containing salts, amino acids, and soluble proteins that nourish organs and transport essential substances throughout the tick’s body [90,91]. Following the heart, paired salivary glands, resembling grape clusters, are antero-laterally situated. Through salivary ducts that connect the chelicerae and the hypostome [86], these glands release substances into the salivarium, a small space inside the mouth. The primary internal organ, the midgut, resembles a sac with numerous lateral diverticula. In unfed ticks, these tube-like diverticula expand during feeding and fill with blood, aiding digestion and influencing tick growth, reproduction, and fitness [92].

Malpighian tubules manage excretion by releasing nitrogenous wastes into the rectal sac, maintaining the tick’s internal balance by eliminating metabolic by-products, mainly in the form of guanine. The reproductive system includes male structures like testes, vasa deferentia, seminal vesicle, and ejaculatory duct, while females have an ovary, oviducts, uterus, vagina, and seminal receptacles. The ovary expands significantly during feeding and mating, containing large, amber-colored eggs in gravid females [70].

### 1.3. Life Cycle of Hard Ticks

The tick life cycle encompasses distinct phases, including host-seeking, feeding, development, metamorphosis, and reproduction, interspersed with diapause (dormancy) periods aligned with seasonal changes [93]. Hard ticks have a single nymphal instar, while soft ticks have two or more nymphal instars. Throughout their life cycle, ticks, as obligate ectoparasites, rely on blood feeding [70], characterized by short feeding phases and extended non-feeding intervals [94]. Sexual dimorphism is evident only in the adult stage [95], and host selection may differ between juvenile and adult ticks of the same species [10]. Over 90% of Ixodid species exhibit a three-host life cycle, with larvae and nymphs feeding on small to medium-sized hosts and adults targeting larger host species [70].

*Ixodes scapularis*, for example, feeds on various host species, showcasing preferences in host selection [82]. The number and variety of hosts for hard-bodied ticks vary, with the completion of the typical three-host life cycle ranging from less than a year to 5 or 6 years under favorable conditions [70]. Most arthropods display a photo-periodically controlled life cycle to synchronize development stages with favorable climatic conditions [95]. Host-seeking diapause commonly occurs during periods of decreasing photoperiod [70], enabling ticks to endure harsh environments and conserve energy. Diapause is crucial for ticks to survive unfavorable conditions, during which they reduce metabolic rates and abstain from feeding [96]. Over 90% of the life cycle occurs off the host, with molting typically taking place in sheltered microhabitats such as soil, leaf litter, or host nests [70,97].

Ticks regulate body water levels by absorbing water vapor from the air, a critical aspect of their survival [98]. The critical equilibrium activity (CEA) represents the minimum moisture level needed for water balance [99]. Ticks have adaptations to minimize water loss, such as a modest metabolic rate, a relatively impermeable cuticle, and the excretion of solid urine in the form of guanine [100]. The careful regulation of water balance is crucial for ticks, influencing their distribution, survival, and activity and playing a critical role in their capacity to transmit disease-causing pathogens [101]. Some Ixodids have evolved as two-host species (e.g., *Hyalomma isaaci*), where larvae attach to a host, ecdyse on the host upon completing feeding, and the resulting nymphs reattach and complete feeding. Single-host tick species, like *Rhipicephalus* (*B*) *microplus*, undergo the larval, nymphal, and adult stages on a single-host animal, feeding and molting during each phase. Upon becoming engorged, the female tick detaches from the host to lay eggs [67,94] (Figure 3).

## 2. Hard Ticks: An Efficient Vector for TBDs

The role of ticks and the pathogens they carry has become an increasingly prominent concern for global human and animal health [56]. Certain ticks employ an ambush strategy, residing in open environments and crawling onto vegetation to await passing hosts, a behavior known as questing [102]. While questing, ticks sway their forelegs, exposing ‘Haller’s organs’, located at the end of the first pair of legs. These organs detect various host odors, humidity, body heat, vibrations, and carbon dioxide [70,102]. Genera like *Rhipicephalus*, *Haemaphysalis*, and *Ixodes* have larvae, nymphs, and adult stages that engage in questing on vegetation. Ticks attach to hosts by utilizing their front legs and then traverse the skin to locate an appropriate site for attachment and feeding. Conversely, adult ticks belonging to the *Amblyomma* and *Hyalomma* genera are proactive hunters, running across the ground after nearby hosts [103].

Ticks play a crucial role in host range evolution, impacting host reproductive success and population dynamics, especially at high infestation intensities [94]. The efficient transmission of pathogens depends on vector competence, which encompasses the ability to acquire, maintain, and transmit pathogens, with factors specific to tick species and associated pathogens [104]. Experiments have shown that different tick species can carry and spread different pathogens. This highlights the need for studies that focus on specific regions and use different tick species [105,106,107,108]. For example, *Ixodes scapularis* is the main tick that spreads POWV II (Powassan (POW) virus lineage II), and *Dermacentor variabilis* and *Amblyomma americanum* ticks in the eastern USA can also get infected and transmit POWV II because they live in the same places and feed on the same hosts [105]. Researchers observed variations in the vector competence of *Rhipicephalus appendiculatus* and *Rhipicephalus zambeziensis* ticks in transmitting *Theileria parva*, which were attributed to differences in parasite stock and tick population [106]. Comparisons between *I. scapularis* and *I. pacificus*, the vectors for LD, highlighted that *I. pacificus* had higher attachment rates and engorgement, while *I. scapularis* transmitted the pathogen more efficiently [104].

Ticks can acquire multiple pathogenic microorganisms, such as bacteria, viruses and protozoans. This occurs through two main mechanisms: systemic transmission and co-feeding [109,110]. The main way that tick borne pathogens (TBPs) are spread is through horizontal (oral) transmission, which means that the pathogen moves from an infected tick to a definitive host that is not infected and back again [111]. Co-feeding transmission enables co-feeding ticks, whether infected or susceptible and potentially at different life stages, to transfer TBPs. This occurs directly, even in the absence of established pathogens within the reservoir host [112] (Figure 4). For example, LD, Thogoto virus, and TBE virus are diseases that can be transmitted from infected ticks to non-infected ticks during close-proximity feeding [113]. In systemic transmission, ticks can maintain pathogens through transstadial, sexual, and transovarial routes [114]. In the systemic transmission pathway, questing ticks obtain viruses by feeding on infected hosts, and during subsequent feeding, infected ticks can transmit the virus to susceptible hosts [115]. The transstadial transmission of pathogens from one developmental stage to the next [116] occurs in *Borrelia*, *Rickettsia*, and *Anaplasma phagocytophilum* [113]. Sexual transmission is observed in *Rickettsia* and some relapsing fevers during copulation [117]. 

Hosts, vectors, and pathogens engage in an ongoing antagonistic arms race, co-evolving to enhance their respective performance and fitness [118]. The route of transmission is a crucial factor shaping the co-evolutionary dynamics between hosts and parasites [118]. The tick microbiome, which comprises commensal and symbiotic obligate endosymbionts, affects the tick’s ability to grow, reproduce, adapt to feeding sources, and protect itself from environmental stresses [119]. Non-pathogenic microorganisms in ticks can influence TBP replication, either inhibiting or enhancing it, and play a crucial role in determining the outcome of TBP transmission in tick populations [120].

### 2.1. Blood-Feeding Mechanism of Hard Ticks 

The tick-feeding process involves intricate mechanisms, including the regulation of specialized structures such as the chelicerae and the pre-oral canal, with labrum elevation, pharyngeal valve action, muscle contractions, and neural signaling [121]. During feeding, the size of larvae, nymphs, and females significantly increases, resulting in engorged individuals adopting a lenticular or egg-shaped form resembling a blood-filled sac. In a non-fed state, hard-bodied ticks exhibit a flat and elongated form, transitioning to an oval or nearly circular shape when engorged [82] (Figure 5).

Female hard ticks need a blood meal to lay eggs and can mate either while on their host (metastriate ixodids) or on or off the host (prostriate ixodids) [122]. Interestingly, adult males show minimal change in body length following feeding [81,123]. Unlike females, adult males rarely engorge over twice their body weight [124]. Male ticks of most Ixodid species display intermittent feeding behavior, with prostriate ticks experiencing spermatogenesis before adult feeding, while metastriate ticks generally require blood feeding to initiate sperm development. However, once stimulated, metastriate males can detach from hosts and actively search for female ticks to mate [8].

Male mouthparts play an active role in reproduction by penetrating the female genital pore and delivering the spermatophore. The relationship between salivary gland components and sexual variances in tick feeding or behavior remains unclear. Tick feeding studies have demonstrated that ticks in different life stages, including larvae, nymphs, and adults, possess the ability to reattach and resume feeding even if their blood meal is interrupted, a phenomenon known as “interrupted feeding” [8].

Ticks exhibit distinctive characteristics compared to other blood-feeding arthropods [125]. They rely on blood for nutrition and reproduction, employing their chelicerae to create blood pools for feeding [126]. Ticks possess a unique feeding mechanism, penetrating the host epidermis and dermal capillaries with their chelicerae and drawing in fluids exuded into the wound. Ticks have varied mouthpart adaptations for different skin penetration depths, with ixodid ticks having longer mouthparts, while argasid ticks possess well-developed chelicerae for swift penetration [5].

Ticks’ mouthparts have a hypostome with backward-pointing barbs, and many ixodid species secrete cement around the hypostome for secure anchoring [127]. The origin of cement appears to be from the cells of both type II acini and type III acini within the salivary glands [128]. The Ixodidae family exhibits attachment cement production as a distinctive trait, in contrast to its apparent absence or rarity in Argasidae. Notably, there are no reports of cement production in *N. namaqua* [128]. Ticks of different genera secrete varying amounts of cement, which affects their attachment to hosts [127]. Ticks in different genera produce varying amounts of cement, impacting their attachment to hosts [127]. The cement, besides its adhesive properties, is proposed to have antimicrobial functions, including sealing lesions during feeding, facilitating feeding and pathogen transmission, and protecting ticks from host immune and inflammatory responses [129].

Feeding duration varies among tick species, with ixodid ticks taking several days to a couple of weeks, while argasid ticks are notably faster, often feeding for less than an hour [23,127]. During a single feeding session, an ixodid tick can consume over 200 times its unfed body weight, significantly increasing the risk of pathogen acquisition from an infected host [27,130]. Ticks have a unique cuticle with properties distinct from other arthropods, using resilin to enhance cuticle flexibility. When ticks get swollen, big changes happen to the epithelial cells in the midgut diverticula, which makes it easier for the ticks to handle changes in volume [22]. Ticks exhibit intracellular digestion, known as heterophagy, unlike many other blood-feeding arthropods. Ticks digest their meal entirely within the epithelial cells of the midgut, allowing them to survive extended periods without feeding and harbor pathogens over time, contributing to their role as disease reservoirs [131,132]. Ticks, with their unique feeding mechanisms, digestion processes, and disease transmission capabilities, are captivating subjects in the fields of parasitology and vector-borne diseases [133].

### 2.2. Specialized Function of Salivary Glands in Hard Ticks 

It is important for ticks to have multifunctional salivary glands (SGs) that help them stay alive and spread diseases. These glands are also used to study development and transmission routes, which has led to possible interventions that target these glands because they are so important to tick survival and vector competency [56,86,97,134,135,136]. The salivary glands of female argasid and ixodid ticks consist of numerous acini, categorized into two types in argasid ticks (types I and II) and three types in ixodid ticks (types I, II, and III). In ixodid males, there is an additional type (type IV) of acini in their salivary glands. In both male and female ticks, Type I acini primarily connect to the anterior portion of the main salivary duct, whereas Type II and III acini are linked to secondary and tertiary ducts located more distally. This organization reflects the specialized functions of these acini in tick salivary gland activity [134,136]. Distinct acini types, such as type I for hydration and type II/III for saliva production, serve specific functions within these glands [137]. Newly hatched larvae exhibit minute salivary glands with only discernible ducts. In a mature larval stage, some alveoli start to form, such as types 1, 2, and 3 alveoli observed in *H. spinigera* larvae. Type 4 alveoli are not distinguishable during the larval stages [109].

Tick saliva serves various functions, including water balance, holdfast and gasket formation, regulation of host responses, dynamics, individuality, mate guarding, and saliva-assisted transmission (SAT) [23,27]. SAT, which facilitates pathogen transmission through arthropod saliva, is well-documented in blood-feeding arthropods, particularly ticks [134,138,139]. SAT greatly enhances co-feeding transmission, an efficient disease transmission strategy employed by ticks [111]. Co-feeding depends on how easily pathogens can be transmitted and how well each host species can support this route. It is affected by the fact that infecting and infection-acquiring instars feed on the same host at the same time, which is linked to tick activity seasons [140]. Host reactions, such as hemostatic plug formation (initial response to vessel wall injury), involve platelets promptly engaging with any breaks in the vascular endothelium [141], and inflammatory responses aim to reject feeding ticks [23,126,134]. However, ticks counteract these defenses using biologically active molecules in their salivary glands [142]. These molecules evolved through host–parasite co-evolution and exhibit anticoagulant, antiplatelet, vasodilatory, anti-inflammatory, and immunomodulatory properties. Tick proteins, such as Prostacyclin 51 and IxscS-1E1 52, disrupt platelet aggregation. Ticks called *I. scapularis* and *Haemaphysalis* release different substances that stop and start the coagulation cascade, which changes how blood clots [13]. Ticks employ these bioactive compounds to facilitate successful feeding, ensuring their survival and reproductive success [130,132,134].

Tick saliva contains an arsenal of bioactive molecules that modulate host hemostasis and immune reactions, thus enabling blood acquisition [143]. A protein with anti-complement properties has been identified, cloned, and expressed in *I. scapularis*. It inhibits C3b binding and expedites the dissociation of factor Bb from the alternative pathway of C3 convertase in the host. The tick saliva and the recombinant protein exhibit comparable inhibitory effects on the complement system [144]. Hard ticks like *R. sanguineus*, *R. appendiculatus*, *Dermacentor reticulatus*, and *Amblyomma variegatum* produce proteins in their salivary glands. These proteins include evasins, which bind to chemokines [133,145]. These evasins bind to host chemokines, hindering their ability to activate chemokine receptors [145], and they serve as chemoattractants for immune cells, influencing leukocyte recruitment and host inflammation. For instance, Evasin-1 binds to CCL3, CCL4, and CCL18, while Evasin-3 binds to CXCL8 and CXCL1. Researchers have used in vitro tests and given recombinant Evasin proteins to BALB/c and C57BL/6 mice to show that they can stop inflammation [133]. The macrophage inhibitory factor (MIF) found in the saliva of *A. americanum* and *H. spinigera* ticks strongly inhibits the recruitment of neutrophils and monocytes to the attachment site [146,147]. Apyrase (ATP-diphosphohydrolase) is an enzyme that plays a role in inhibiting platelet aggregation by converting the active forms of ATP and ADP into the inactive form AMP [148] (Figure 6).

Interestingly, the innate immune responses in arthropods are predominantly coordinated through the Toll, immune deficiency (Imd), and Janus kinase (JAK)-signal transducer and activator of transcription (STAT) pathways [149]. The tick JAK/STAT pathway responds to host-derived interferon gamma during feeding, activating antimicrobial effectors like Dae2 and the 5.3-kDa AMP, which limit the proliferation of pathogens such as *Borrelia burgdorferi* and *Anaplasma phagocytophilum.* Peritrophin-1 is a tick JAK/STAT effector that plays two roles; namely, it changes the integrity of the peritrophic matrix to help *B. burgdorferi* survive in the gut and stop *A. phagocytophilum* from colonizing. The JAK/STAT pathway in ticks complicates the balance between positive and negative regulation of infection, highlighting its complex role in responding to TBPs [140].

## 3. Distribution and Diversity of Tick Vectors

The hard ticks exhibit diverse distribution and significance across continents. In America, prevalent hard tick genera infesting pets include *Amblyomma*, *Dermacentor*, *Ixodes*, *Rhipicephalus*, and the recently established *Haemaphysalis longicornis* [123,127]. Australia faces economic losses and disease transmission primarily from *Ixodes*, *Haemaphysalis*, and *Rhipicephalus* [150,151]. Europe, Northern Africa, and Southern Africa collectively report a total of 67 tick species, which includes the genera: *Amblyomma*, *Dermacentor*, *Haemaphysalis*, *Hyalomma*, *Ixodes*, and *Rhipicephalus* [152,153]. Avian hosts in Europe harbor 37 hard tick species, exhibiting preferences for specific hosts, such as seabirds in Western–Northern Europe (*Ixodes rothschildi*, *I. unicavatus*, and *I. uriae*) or associations with tortoises (*Hyalomma aegyptium*) and birds of prey (*Rhipicephalus turanicus*, *R. sanguineus*) [154]. In China, the Ixodidae family comprises 111 species distributed across seven genera: *Amblyomma* (8 species), *Anomalohimalaya* (2 species), *Dermacentor* (14 species), *Haemaphysalis* (43 species), *Hyalomma* (7 species), *Ixodes* (29 species), and *Rhipicephalus* (8 species) [155].

### 3.1. Distribution of Hard Ticks in India

Over the years, our knowledge of ticks in India has evolved through various contributions, including those of [156,157,158,159]. The compilation by the National Institute of Virology and other institutes includes 106 valid tick species collected from different hosts across the country [160].

Between 1997 and 2023, studies in India significantly advanced our knowledge of tick populations, revealing their diverse presence and distribution. Researchers in India have identified nine genera of hard ticks, which include *Amblyomma*, *Aponomma*, *Haemaphysalis*, *Hyalomma*, *Ixodes*, *Rhipicephalus*, *Boophilus*, *Dermacentor*, and *Nosomma* [160,161]. Kerala, for instance, has reported various tick species on domestic animals [162,163,164,165]. The species of *Haemaphysalis*, *Amblyomma*, *Ixodes*, and *Rhipicephalus* were documented in the Western Ghats [166].

Researchers have identified several tick species in Tamil Nadu, including *Amblyomma integrum*, *Haemaphysalis bispinosa*, and *R. turanicus*, among others [65,167,168]. Andhra Pradesh reports eight tick types, including *Rhipicephalus* (*B*) *microplus* and *Hyalomma marginatum* [169,170]. Other states like Chhattisgarh, Odisha, Maharashtra, Gujarat, Rajasthan, and Uttar Pradesh also document diverse tick genera and species [171,172,173,174,175,176,177]. This extensive literature underscores the critical importance of studying ticks and TBDs in India, offering valuable insights into their distribution and diversity (Figure 7). Table 1 provides an overview of important tick-borne diseases and their respective vector distributions in India.

### 3.2. Genetic Diversity of Ticks

Studies in population genetics related to wingless arthropod vectors, like ticks, have provided valuable insights into their dispersal patterns. These patterns are closely tied to host mobility and behaviors, significantly influencing genetic structure and disease transmission dynamics [178]. Researchers utilized molecular markers to address questions about genetic variability, population genetic structure, gene flow, and potential genetic isolation by distance in one or more tick species [59,74,178]. Additionally, newer studies have employed whole-genome sequencing and functional genomics, recognizing the potential of next-generation approaches in advancing our understanding of tick biology and evolution [178,179].

Understanding genetic diversity, stemming from factors like DNA recombination, mutations, gene flow, and genetic drift, is crucial [180]. Researchers commonly use mitochondrial genomes in molecular systematics because they undergo faster evolutionary changes [181]. In ticks, the genomes of 66 species from 18 genera have been sequenced, offering consistent genus-level classification [182]. 

The impact of host mobility on the genetic composition of tick populations is variable. Studies on *I. scapularis* and *Ornithodoros coriaceus* reported limited gene flow despite their hosts’ high mobility [178]. In contrast, ticks associated with less mobile hosts exhibit reduced gene flow, while those connected to highly mobile hosts show higher gene flow. For example, *A. americanum* and *A. triste* displayed increased gene flow attributed to the dispersal capabilities of their hosts in Arkansas ecoregions, Georgia, and Argentina [183]. The genetic diversity of *Rhipicephalus sanguineus s.l.* ticks in Colombia, which are known to be relevant in TBD transmission, is important for a complete study of tick-borne disease epidemiology [184].

High genetic diversity and low differentiation observed in *Rhicephalus* (*B*) *microplus* populations suggest an infinite island model in migration-drift equilibrium in the provinces of Zimbabwe [185]. Gene flow through cattle movement was noted, leading to allele loss and founder effects (refers to the decrease in genomic variability that arises when a small group of individuals becomes isolated or separated from a larger population during migration), potentially impacting tick adaptation in diverse habitats [186]. Host species influence the genetic structure of *A. dissimile* ticks in a 500 km area, resulting in the formation of small breeding groups [187]. Host mobility during immature stages contributes to the distinct genetic structures observed in two Ixodid ticks, *Haemaphysalis flava* and *Ixodes ovatus*, in Niigata Prefecture, Japan. *I. ovatus* has a stronger genetic structure because its small mammal host does not move around much, while *H. flava* does not have any genetic structure because its host probably moves around a lot with birds [188].

A phylogenetic tree generated by the 16S mitochondrial gene from *Rhipicephalus sanguineus* available in the database showed mixed genetic patterns. We observed two prominent groups. Group I consisted of ticks from South America, Africa, and Asia, and Group II predominantly represented ticks from European regions (Figure 8). However, population structure insights on important tick vectors across different geographic locations are essential for assessing disease transmission dynamics, host adaptation, and overall tick ecology.

Hybridization, a natural phenomenon, plays a pivotal role in the emergence of new species on Earth [189]. In nature, ticks engage in hybridization activities, supported by both observational evidence and experimental studies. Researchers have documented natural hybridization between *Ixodes ricinus* and *I. persulcatus* ticks in Estonia [190]. Furthermore, laboratory studies have reported hybridization in ticks of the genus *Ixodes* for several species [191]. A recent study that looked at how the tick-borne encephalitis virus (TBEV) is spread in *I. ricinus* and *I. persulcatus* ticks, including hybrids, found that many complicated factors affect how TBEV is spread [192].

## 4. Impacts of Climate Change on the Expansion of Hard Ticks

Climate change has an impact on tick populations since it can modify their habitat and areas where ticks can persist over extreme weather conditions. Ticks exhibit diverse distribution patterns globally, which are influenced by climatic factors like temperature, humidity, altitude, vegetation types, and host availability [193,194,195,196]. Climate change, especially the rapid rise in global temperatures, has played a significant role in the expanded range of arthropod vectors, including ticks [140,197,198,199]. Recently, a warmer climate in the northeastern USA has been linked to the geographical expansion of an aggressive human-biting tick *A. americanum*, commonly known as the lone star tick, impacting its range and population dynamics, and presenting a growing public health challenge [200]. Also, warmer climates are associated with accelerating the maturation rates of *I. scapularis* ticks which may likely elevate the prevalence of LD in the northeastern United States by extending the transmission season [201]. Projected future climatic conditions in Europe are expected to increase the suitability of habitats for *Ixodes ricinus*, *D. reticulatus*, and *Dermacentor marginatus*, resulting in expanded geographical ranges, especially towards Eastern Europe [202]. In Russia, climate changes are responsible for the expansion of *I. persulcatus* toward newer areas like Russia’s Arkhangelsk Oblast (AO) and Komi Republic [203]. In China, *I. ovatus* is distributed mainly in southwest and northwest China, but changes in the regional temperature profile and annual precipitation may influence its habitat distribution, with a predicted expansion toward northeast China in the future [204]. Climate change, particularly rising temperatures, may adversely affect tick species in tropical zones, forcing some to colonize new areas and leading to changes in habitat suitability, such as the dispersion of ticks like *Amblyomma variegatum* into regions beyond those with prolonged dry periods, as observed in Zimbabwe [197].

Ticks have evolved sophisticated life-history strategies, including behavioral and physiological adaptations, to maximize host exploitation [197,198]. Climate change’s indirect impacts on host communities might promote the increased transmission of TBPs through alterations in tick abundance [201,205]. Understanding factors like host specificity, habitat suitability, and climate tolerance is crucial for comprehending tick range expansion [200]. Climate and host availability play a crucial role in shaping tick life cycles and feeding physiology, leading to their shift from predatory to ectoparasitic lifestyles [5]. An increase in the availability of small mammal host density correlates with heightened questing nymph and adult tick densities, reflecting accelerated progression through larval and nymphal stages [198,199,206] (Figure 9). 

Host–parasite associations, particularly host-specificity relationships, play a role in constraining many tick species [207]. A variety of vertebrates act as reservoirs for tick-borne pathogens, contributing to the persistent nature of these diseases and posing risks to domestic animals and humans [201,208]. Understanding the relationship between host diversity and disease risk is complex, with global trends suggesting that higher host biodiversity can increase tick exposure. However, local effects may vary, and introducing highly competent hosts can raise transmission risk [94]. Wildlife vertebrate hosts play a crucial role in the ongoing cycles of TBPs. In some instances, they have been significant factors in the recent surge of ticks and tick-borne illnesses [30,209]. Various vertebrates, including lizards, birds, hedgehogs, snakes, and mammals, serve as reservoirs for pathogens; for example, wild boars are known reservoirs for many tick-borne viruses, bacteria, and parasites, which play a crucial role in the sylvatic cycle [210,211,212]. Rodents from various families serve as zoonotic reservoirs for ectoparasites like mites and ticks, playing a vital role in the transmission of various vector-borne diseases such as scrub typhus, and TBDs [210]. Wild animals, including species like wild boars, lizards, and snakes, commonly act as extensive and frequently undiscovered reservoirs of hosts for zoonotic diseases, particularly those transmitted by ticks [212,213]. 

Both large-wildlife loss and climatic changes can independently influence the prevalence and distribution of zoonotic disease. The decline of large-wildlife populations globally disrupts ecological functions like disease control, potentially impacting tick abundance and tick-borne disease risk due to altered host dynamics and vegetation structure. Understanding the interplay between host loss, vegetation changes, rainfall and tick survival is crucial for assessing the overall impact on vector-borne disease transmission [214,215]. In India, studies on the impact of climate change on the expansion of Ixodid ticks have not been documented.

## 5. Important TBDs in India Transmitted by Hard Ticks 

In India, TBDs pose a significant public health concern, presenting unique scenarios such as KFD [216,217], CCHF [47,218,219], rickettsial infections like Indian Tick Typhus (ITT) [220], LD [221], Q-Fever [222], and anaplasmosis [223].

### 5.1. Kyasanur Forest Disease (KFD)

In 1957, researchers first identified KFD in the Kyasanur forest of India. Caused by the KFDV, a member of the Flavivirus family, it can induce hemorrhagic illness in various animals [224]. The primary mode of transmission is through ticks, particularly *H. spinigera* [225,226,227]. Common symptoms include chills, headaches, muscle aches, vomiting, bleeding, and high fever, with a mortality rate ranging from 3–5% [41,46,216,217].

Ticks, prevalent in regions characterized by drier and warmer climates, exhibit a 58.66% prevalence in Karnataka (Shivamogga, Chamrajnagar, and Chikmagalur districts), 12.02% in Kerala (Kozhikode and Wayanad districts), 18.15% in Maharashtra (Raigad and Ratnagiri districts), and in the Nilgiris districts of Tamil Nadu and North Goa district in Goa. Nymphal activity occurs during the non-rainy season [226].

Ticks, mammals, and birds maintain the KFD virus through cycles. Monkeys, particularly *Macaca radiata* (bonnet macaque) and langurs in the genus *Semnopithecus*, are susceptible and often fatal hosts, yet they serve as important reservoirs. Cattle, the primary hosts for the ticks, do not amplify the virus, and their role in KFD transmission requires further investigation [227,228]. Tick species such as *Haemaphysalis turturis*, *Haemaphysalis kinneari*, *Haemaphysalis kyasanurensis*, *Haemaphysalis wellingtoni*, *Haemaphysalis minuta*, *Haemaphysalis cuspidata*, *Ixodes petauristae*, *Ixodes sceylonensis*, *Dermacentor auratus*, and *Rhipicephalus haemaphysaloides* have been reported in eleven out of 29 states and 7 union territories in India, and they are capable of carrying the virus [45,229].

### 5.2. Crimean–Congo Hemorrhagic Fever (CCHF)

CCHF virus, belonging to the Nairovirus genus within the Bunyaviridae family, is responsible for causing severe illness in humans [230]. Since its initial identification during a 1944 outbreak in the West Crimean region of the former Soviet Union and official isolation in 1956, CCHF has been documented in various regions, including parts of Africa, Asia, Eastern Europe, and the Middle East [47]. CCHF is a tick-borne viral disease with an average mortality rate ranging from 30% to 50% [47,49,231].

During a nosocomial outbreak in 2011, CCHF was first confirmed in India, specifically in Gujarat State. Subsequently, Gujarat has experienced numerous outbreaks and sporadic cases of CCHF [49]. In 2014 and 2015, a nosocomial outbreak was recorded in a private hospital in the neighboring state of Rajasthan. The virus was detected in the southern Indian state of Kerala in 2016, associated with the travel history of a slaughterhouse worker who came from Oman [218]. In 2019, there was an outbreak in Gujarat that reportedly had a mortality rate exceeding 50% [219].

Domestic animals and *Hyalomma* ticks are involved in amplifying and transmitting the CCHF virus, although the virus has also been isolated from *Rhipicephalus*, *Boophilus*, and *Dermacentor* species, which may serve as additional vectors [232]. Various factors such as land use changes and climate have been identified as crucial drivers of CCHF infections [218]. The major routes of CCHF transmission to humans include nosocomial infections, tick bites, crushing ticks with bare hands, and contact with the blood or tissue fluids of infected animals or humans. Available information suggests that CCHF cases mainly result from occupational exposure among abattoir workers, farmers, veterinarians, and healthcare workers [219]. A wide range of vertebrates, such as cattle, goats, donkeys, and horses, along with smaller wildlife species like hares and hedgehogs, act as reservoirs for the CCHF virus [233,234].

### 5.3. Lyme Disease (LD)

In 1976, LD garnered attention when a cluster of juvenile arthritis cases in Old Lyme, Connecticut, accompanied by skin lesions resembling those from European tick bites, raised suspicions of a shared infectious agent [235]. LD is primarily caused by the *B. burgdorferi* spirochete. The increasing risk of LD in the United States is mainly attributed to the expanding range of the blacklegged tick, *I. scapularis*, the principal vector for the spirochetal pathogen *B. burgdorferi* [236]. It predominantly affects the skin, nervous system, heart, and joints [221,236]. In Lyme borreliosis, the infection process involves adaptations for survival and replication in the mammalian host, evading host immune mechanisms [237,238]. A literature search revealed a rare cluster of five LD cases over three months in Rohtak, Haryana, a non-endemic region of India. The patients, including a farmer, a boy, and housewives, presented with distinct skin manifestations following tick or insect bites, with serological tests confirming *B. burgdorferi* infection. These cases highlight an unusual occurrence in a region not traditionally associated with LD [40]. A study in North India involving 252 individuals found 18 cases of LD. They used a standard two-tiered testing algorithm (STTA) with ELISA and immunoblot assays. Intrathecal IgG synthesis indicated a 7.14% antibody response to *B. burgdorferi* infection. Molecular evidence showed *B. burgdorferi* sensu lato and *Anaplasma phagocytophilum* in neuroborreliosis patients [221].

### 5.4. Tick-Associated Rickettsial Pathogens 

Rickettsiae are intracellular endosymbionts. They are spread by blood-sucking ectoparasites like ticks, fleas, and lice [239]. Approximately 24% of terrestrial arthropods carry Rickettsia endosymbionts [240]. Infections caused by Gram-negative bacteria like *Rickettsia*, *Orientia*, *Ehrlichia*, *Neorickettsia*, *Neoehrlichia*, and *Anaplasma* are the common pathogens involved in typhus and spotted fever groups [241,242].

The dog tick *R. sanguineus* is the primary vector for ITT (Indian Tick Typhus) in India, although certain species of *Haemaphysalis* and *Hyalomma* ticks can also transmit the infection. Other than scrub typhus (non-scrub typhus rickettsial diseases), rickettsial diseases, including spotted fever group (SFG) rickettsioses and typhus group (TG) rickettsioses, are not uncommon in India [243]. Delhi, Himachal Pradesh, Jammu and Kashmir, Uttarakhand, Rajasthan, Tamil Nadu, and Maharashtra have reported several cases. In the early years, cases were mainly reported from soldiers on the Assam and Burma fronts during the Second World War, when it was a problem second only to malaria [244]. Globally, diverse SFG rickettsioses occur, highlighting the need for regional understanding and control measures [245].

### 5.5. Eco-Epidemiology of TBDs 

The dynamics of TBPs, encompassing their occurrence and abundance, are influenced by ecological interactions between tick species and their vertebrate hosts [246]. The expansion of tick species intricately links to the complex dynamics of TBPs, which rely on the abundance and distribution of hosts that sustain tick populations and act as reservoirs for these pathogens through intricate demographic processes and movement dynamics [247] (Figure 8).

Ecological epidemiology explores the interactions between hosts and their pathogens at the population and community levels, encompassing infectious disease studies in both human and wildlife populations [248,249]. Many human diseases transmitted by arthropods, insects, or other invertebrates or vertebrates originate in animals [21]. Alterations to ecosystems through infrastructure development, such as deforestation and converting forestlands for urbanization, contribute to changes in disease epidemiology [250]. Diseases often emerge from pathogens sustained in complex natural transmission cycles involving interacting host and vector communities. The composition of these communities significantly influences pathogen amplification and the risk of spillover transmission [251]. The complicated eco-epidemiology of TBP transmission cycles is mostly because tick vectors have a life cycle that changes and adapts. Climate change also affects arthropod populations and the dynamics of reservoir hosts [252] (Figure 9).

Dynamic interactions among biotic and abiotic elements shape the epidemiology and ecology of TBDs [253]. The pathogens that they carry or the disease they transmit may affect their distribution areas and their population densities. These changes can be caused by climate change, human activities, land use, economic and political factors, and changes that happen naturally in ticks and the diseases they carry [254].

KFD virus was limited to Karnataka’s Shimoga, Chikmagalur, Uttara Kannada, Dakshina Kannada, and Udupi districts in India [255]. Since 2012, the virus has extended to Nilgiris in Tamil Nadu, and cases were reported in Kerala, Goa, and Maharashtra from 2013 to 2017, raising concerns due to the shared Western Ghats ecology in these regions [44]. The potential transmission of KFD poses a significant threat to residents and workers in forested areas in these regions. The movement of monkeys and rodents further compounds the risk, serving as hosts for vector ticks perpetuating the KFD virus in the natural environment [44,255]. The intricate eco-epidemiology of TBP transmission, exemplified by the CCHF virus, demands a nuanced understanding of tick vectors, climate influences, and reservoir host dynamics.

### 5.6. Molecular Diagnostics and Emerging TBDs

The effective management of TBDs is very important and requires efficient diagnostic tools for correct treatment [256]. Serology-based diagnostics have been used for decades, but initial non-specific clinical presentations and the delayed appearance of specific antibodies (15–26 days) for serological diagnosis can mask the identification of TBDs, hindering timely and impactful confirmation by specific laboratory testing [257]. The Weil–Felix Test (WFT), Latex Agglutination Assays (LAA), and Western blots commonly used for the diagnosis of TBDs are low in specificity and sensitivity. Furthermore, ELISA, a widely used diagnostic assay, struggles with varying specificity and sensitivity; for example, an FDA-licensed ELISA for anti-IgM *Rickettsia typhi* exhibits 45% sensitivity and 98.3% specificity [258,259]. The cross-reactivity of antibodies between different pathogens or strains can lead to false-positive test results, further complicating the situation [260].

In recent times, the use of molecular diagnostics for TBPs has shown many advantages as they are technically straightforward, more specific, and sensitive [260,261]. Different types of PCRs (polymerase chain reactions) have been used for the clinical diagnosis of TBPs [262]. Conventional PCR involves amplifying target genes and visualizing it on an agarose gel; it can be further sequenced for phylogenetic analysis of the pathogens [261,262]. Recent real-time (RT) PCR methods, employing chemistries like SYBR Green (binds specifically to double-stranded DNA) and TaqMan (fluorogenic probe for specific PCR product detection during amplification cycles), offer advantages such as the quantification of pathogens, high sensitivity, and specificity [263,264]. TaqMan-based multiplex RT PCRs are another notable development that allows for the detection of multiple pathogens in single-tube reactions [265]. For example, the newly developed Tick Path Layerplex assay employs RTPCR technology to simultaneously detect and characterize 11 pathogens responsible for these diseases in domestic dogs, and it shows high sensitivity and compatibility with standard diagnostic instruments [266]. When used with serologic testing, whole blood real-time polymerase chain reaction (WB-RTPCR) improves the ability to diagnose early LD (ELD) [267]. Multi-locus sequence typing (MLST) enhances genotypic characterization for population studies [268]. For example, MLST identified nine *Rickettsia* species, including six human pathogens, from tick DNA samples, highlighting diverse rickettsial species diversity [269]. Next-generation sequencing (NGS) has greatly advanced our knowledge of tick-borne pathogens and their impact on human health, uncovering new microbes and providing insights into tick microbiome diversity. These studies also contribute to understanding evolutionary relationships, identifying diagnostic targets, and innovating approaches to tackle tick-borne diseases (TBDs) [270].

## 6. Challenges Associated with the TBDs

Managing ticks and TBDs poses a significant challenge due to variations in diagnostics, treatment, disease surveillance, control, and management strategies [271,272,273]. Ticks, acting as vectors and reservoirs of pathogens, cause various diseases in humans and animals and are responsible for substantial harm to livestock [26]. Apart from causing diseases in humans and animals, tick bites can trigger severe allergic reactions in humans [28,274]. For example, alpha-gal syndrome is a rare allergic reaction to a sugar molecule called alpha-gal, injected through tick bites, leading to the delayed onset of allergic symptoms triggered by the consumption of red meat [275]. Additionally, these vectors have indirect repercussions on livestock, particularly affecting the dairy industry. The costly and singular focus of diagnostic methods discourages cattle herders from screening for these infections [276]. Controlling ticks in domestic animals to prevent disease transmission, avoid tick paralysis or toxicosis, and mitigate the physical damage caused by ticks is emphasized in India, which has the world’s largest population of cattle at 192.52 million and buffaloes at 109.85 million [52,277].

### 6.1. Strategies to Control TBDs

With the rise in TBD cases and the spread of pathogens to new regions, the demand for effective tick management strategies becomes crucial [277]. Integrated tick management encompasses components such as public education, routine inspection and surveillance, disease diagnostics, and the application of control measures [29]. It combines various methods to minimize the reliance on acaricide treatments. Alternatives include employing anti-tick vaccines, implementing pasture management (which is the practice of growing healthy forage grasses and legumes that ensure a lasting food source for livestock) [278], and utilizing tick-resistant cattle breeds [279]. To enhance control efforts, there is a necessity for Integrated Tick Management (ITM) on larger scales, employing diverse strategies to impact tick populations in residential areas, greenspaces, and public lands [280,281]. Currently, tick population suppression relies primarily on individual homeowners, lacking coordinated efforts at neighborhood or community scales [29]. The effective management of ticks and tick-borne diseases requires the integration of various rational tactics, encompassing biological, chemical, physical, and vaccine technologies on and off hosts [282]. Crucial to this is the safer application of acaricides, considering environmental concerns. Area-wide tick management research should prioritize understanding tick biology, ecology, and behavior, adopting a One Health approach that encompasses human, animal, and environmental health [283].

### 6.2. Acaricides and Biological Control 

Different methods exist for tick control, each with its own limitations [52]. Acaricides are chemicals that are used to get rid of acari. These chemicals are grouped by where they come from or what they are composed of, and they include plant extracts, organochlorines, organophosphates, carbamates, formamidines, synthetic pyrethroids, and macrocyclic lactones [52,284]. Effective tick control primarily occurs on the host, especially for multi-host ticks that spend the majority of their time off the host, with brief feeding periods lasting between four and ten days [285]. Active compounds such as azadirachtin, carvacrol, linalool, geraniol, and citronellal suggest that plant-based compounds could be useful for farmers in rural areas who raise animals for food [286]. In the 20th century, scientists studied tick biocontrol agents like pathogens, parasitoids, and predators. This led to a switch from chemical acaricides to biological control because of health and environmental concerns, cost, and rising pesticide resistance [287]. Classical biocontrol introduces non-native agents to control pests, while augmentative biocontrol uses native predators or parasites. Potential natural enemies for tick control include insectivorous birds, parasitoid wasps, nematodes, bacteria, and fungi, with entomopathogenic fungi being a promising option [286,287,288]. An often-overlooked method involves the careful examination of pets for attached ticks and the removal and destruction of each tick, providing a simple yet effective means of tick control [289]. In summary, a comprehensive approach to tick control involves balancing the efficacy of chemical interventions with the environmentally friendly and sustainable aspects of biological control, recognizing the limitations of each method.

### 6.3. Acaricide Resistance and Mechanism

There are more reports of resistant tick populations around the world [284,290,291]. This is a big problem because it means that tick populations that are exposed to acaricides are passing on traits that make them resistant. Common acaricides target the tick nervous system through various mechanisms, including antagonism, inhibition, modulation, and activation [284,290,291]. Notably, there is a big rise in populations that are not killed by acaricides, mainly ticks like *R.* (*B*) *microplus*, which shows that more research is needed [290]. The study of acaricide resistance mechanisms focuses on target-site insensitivity. It has been found that changes in important nerve function channels, like voltage-gated Na^+^ and K^+^ channels, cause resistance [291]. Metabolic detoxification, which includes more esterases, cytochrome P450, and glutathione S-transferase, seems to be the main way that acaricide-resistant ticks stay healthy [292,293]. Also, ABC transporters, especially P-glycoproteins, are linked to ticks’ multidrug resistance, which shows how complicated acaricide resistance mechanisms are [291].

From a historical point of view, acaricide resistance is a trait that is passed through the vertical route of transmission and changes over time, causing phenotypic resistance, tolerance, and cross-resistance between acaricides that are chemically similar [284]. The study stresses how important it is to keep an eye on field tick populations to see if they are becoming resistant. It suggests a number of tests, such as the larval packet test (LPT), the larval immersion test (LIT), the adult immersion test (AIT), and the larval tarsal test (LTT), which can be used to perform a correct diagnosis [284,291,294]. A case study along the Texas–Mexico border underscores the significance of vigilant monitoring and rapid detection of acaricide resistance. The systematic program, employing the LPT, identified resistance to coumaphos and growing concerns about permethrin resistance. Genetic studies showed different types of fever ticks that came from different places and could infest again. This shows how important genetic tools are for making quick decisions about tick control programs [295].

### 6.4. Drugs/Vaccines against TBDs

The global incidence of tick-borne diseases is increasing each year [26]. Beta-lactam and tetracycline antibiotics, such as doxycycline, have proven effective in treating Lyme disease, Rocky Mountain spotted fever (RMSF), and human ehrlichiosis [296]. In endemic areas, doxycycline is preferred for its reduced frequency of administration and lower adverse effects due to the rising prevalence of coinfections involving *B. burgdorferi* and the human granulocytic ehrlichiosis agent [297]. Ivermectin is a macrocyclic lactone endectocide that works very well against both endo- and ecto-parasites and has antiparasitic effects that last for a long time. Its use, particularly in long-acting formulations like 3.15%, is widespread for controlling *R.* (*B*) *microplus* infestations in cattle, and oxytetracycline was the predominant drug used on farms in various regions [298]. In Brazil, research on the long-term and therapeutic effects of doramectin showed that it effectively got rid of *R.* (*B) microplus* infestations in cattle and kept them from coming back [299].

Vaccines, using the “isolate-inactivate-inject” method, offer a cost-effective strategy against tick-borne diseases [300,301]. Initially using live or killed pathogens, second-generation vaccines with purified components have emerged with technological advancements [301]. In Europe, widely utilized TBE vaccines (formalin-inactivated) include FSME-IMMUN and Encepur, while Russian alternatives such as IPVE and EnceVir are also available for protection against TBEV [302,303,304].

In India, formalin-inactivated KFDV vaccines have been used for decades [305,306]. Currently, there is no definitive treatment for KFD, underscoring the importance of vaccination as a crucial public health measure to manage the disease [305]. The live-attenuated VSV-based KFDV vaccine shows promise, demonstrating safety, eliciting robust immune responses, and providing effective protection in both mouse and macaque models, but it is not yet available for humans [307].

In cattle, anti-tick vaccine (immunological formulations designed to stimulate the host’s immune system, providing protection against tick infestations and the transmission of associated pathogens [308]) success primarily centers around Bm86-based vaccines like GavacTM; however, challenges are posed by tick genetic diversity [308,309]. Subolesin-based vaccines offer promising protection against multiple tick species with efficacy of 80–97%, comparable to or surpassing other tick antigens like Bm86/Bm95, metalloprotease, ribosomal protein P0, ferritin 2, and aquaporin [310].

**Table 1 pathogens-13-00556-t001:** Important TBDs and distribution of vectors in India.

Diseases	Tick Species	Distribution	Author
Kyasanur Forest Disease (KFD)	*Haemaphysalis spinigera*	Karnataka, Tamil Nadu, Kerala, Andhra Pradesh, Orissa, Madhya Pradesh, Meghalaya, Goa, Maharashtra, Gujarat, West Bengal, Andaman and Nicobar Islands	[45,217,224,229,311]
	*Haemaphysalis turturis*	Karnataka, Tamil Nadu, Kerala and Uttar Pradesh	[45,217,224,229,311]
	*Haemaphysalis wellingtoni*	Karnataka, Orissa, Andaman and Nicobar Islands and West Bengal	[45,224,311]
	*Haemaphysalis cuspidata*	Sri Lanka and Karnataka	[224,229]
	*Haemaphysalis aculeata*	Karnataka	[229,311]
	*Haemaphysalis bispinosa*	Andhra Pradesh, Arunachal Pradesh, Assam, Bihar, Goa, Gujarat, Himachal Pradesh, Jammu and Kashmir, Karnataka, Madhya Pradesh, Maharashtra, Mizoram, Orissa, Punja, Sikkim, Tamil Nadu, West Bengal	[160,224,311]
	*Haemaphysalis kyasanurensis*	Karnataka	[45,224,311]
	*Haemaphysalis minuta*	Himachal Pradesh, Karnataka, Orissa and Uttar Pradesh	[45,311]
	*Haemaphysalis kinneri*	Karnataka and West Bengal	[45,311]
	*Haemaphysalis papuanakinneari*	Karnataka and other parts of South East Asia	[311]
	*Ixodes petauristae*	Karnataka	[45,225]
	*Ixodesceylonensis*	Karnataka and Tamil Nadu	[45,225]
	*Rhipicephalus haemaphysaloides*	Throughout India	[45,225]
	*Dermacentor auratus*	Arunachal Pradesh, Assam, Bihar, Himachal Pradesh, Jammu and Kashmir, Orissa, Karnataka, Madhya Pradesh, Maharashtra, Punjab, Uttar Pradesh and West Bengal	[45,225]
	*Hyalomma marginatum isaaci*	Arunachal Pradesh, Bihar, Delhi, Gujarat, Himachal Pradesh, Jammu and Kashmir, Karnataka, Maharashtra, Madhya Pradesh, Orissa, Punjab and Uttar Pradesh	[225]
	*Ornithodorus*	Andra Pradesh, Jammu and Kashmir, Himachal Pradesh, Uttar Pradesh, Karnataka, Maharashtra, Madhya Pradesh, Tamil Nadu and Pondicherry	[225,312]
Crimean-Congo haemorrhagic fever (CCHF)	*Hyalomma anatolicum*	Andhra Pradesh, Assam, Delhi, Gujarat, Haryana, Himachal Pradesh, Jammu and Kashmir, Karnataka, Maharashtra, Madhya Pradesh, Orissa, Punjab and Rajasthan	[17,233,312,313]
	*H. marginatum*	Arunachal Pradesh, Bihar, Delhi, Gujarat, Himachal Pradesh, Jammu and Kashmir, Karnataka, Maharashtra, Madhya Pradesh, Orissa, Punjab and Uttar Pradesh	[17,38,160,312,313]
	*Hyalomma asiaticum*	_	[17,38,160,233]
	*Hyalomma truncatum*	_	[38,160,233]
	*Hyalomma rufipes*	_	[38,160,233]
	*Hyalomma lusitanicum*	_	[233]
	*Hyalomma excavatum*	_	[233]
	*Hyalomma dromedarii*	Andhra Pradesh, Gujarat, Himachal Pradesh, Jammu and kahmir, Maharashtra, Orissa, Punjab and Uttar Pradesh	[17,38,313]
	*Rhipicephalus sanguineus*	Throughout India	[17,160,233,313]
	*Ixodes ricinus*	Northeastern States of India	[31,163]
Indian Tick Typhus (ITT)	*R. sanguineus*	Throughout India	[38,61,242,314]
	*Ixodes ricinus*	Northeastern States of India	[38,314]
	*Haemaphysalis indica*	Bihar, Gujarat, Himachal Pradesh, Jammu and Kashmir, Karnataka, Maharashtra, Orissa, Rajasthan, Uttar Pradesh and West Bengal	[244]
	*H. kinneri*	Karnataka and West Bengal	[244]
	*H. spinigera*	Karnataka, Tamil Nadu, Kerala, Andhra Pradesh, Orissa, Madhya Pradesh, Meghalaya, Goa, Maharashtra, Gujarat, West Bengal, Andaman and Nicobar Islands	[244]
	*H. turturis*	Karnataka, Tamil Nadu, Kerala and Uttar Pradesh	[244]
	*Haemaphysalis leachi*	Eastern West Pakistan, India, and Ceylon, and the terai of southern Nepal	[244]

## 7. Conclusions 

This comprehensive review explores various facets of tick biology, delving into morphology, blood-feeding mechanisms, and the profound impact of salivary glands on host immunity. By dissecting distribution patterns and life cycles, the review provides valuable insights into the intricate ecological dynamics governing tick populations. Given that ticks serve as vectors for a range of diseases, understanding their biology is crucial for public health reasons. The discussion on important tick-borne diseases emphasizes the urgent need for effective control measures. The multifaceted approach advocated for in this review goes beyond tick control, encompassing advancements in vaccine development and population genetics. It calls for progress in vaccine development, recognizing immunization as a pivotal strategy against tick-borne diseases. Simultaneously, a nuanced understanding of population genetics is crucial for predicting adaptability to environmental changes and devising targeted interventions. The review underscores the significance of population genetics not only as an academic pursuit but as a practical tool for predicting and influencing tick population behavior. Geographical and range expansions, influenced by factors like climatic variations, highlight the urgency of anticipating and mitigating the spread of ticks and associated diseases. In essence, this review underscores the interdependence of biological, ecological, and genetic factors in the intricate tapestry of tick-related dynamics. The call for a holistic and interdisciplinary approach reflects the acknowledgment that addressing the challenges posed by ticks and tick-borne diseases requires a comprehensive understanding. This approach not only tackles current issues but also lays the groundwork for anticipating and adapting to the evolving landscape of ticks and associated diseases. The synthesis of these diverse elements encapsulates the complexity of the task at hand and emphasizes the need for collaborative efforts across various scientific disciplines to effectively address this public health concern. This review serves as a valuable resource for researchers and individuals seeking comprehensive information about tick-borne diseases and their evolutionary nature in the context of a constantly changing climate environment, often linked to human actions.

## Figures and Tables

**Figure 1 pathogens-13-00556-f001:**
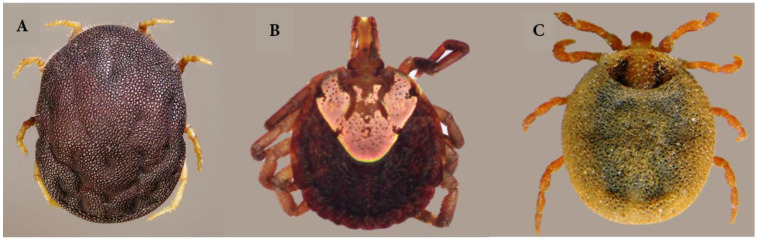
Images of characteristic ticks: (**A**) *Ornithodoros* spp. (Soft tick), (**B**) *Amblyomma integrum* (Hard tick) and (**C**) *Nuttaliella namaqua* (Monotypic tick), (Images (**A**,**C**) adapted from [63,66]), respectively.

**Figure 2 pathogens-13-00556-f002:**
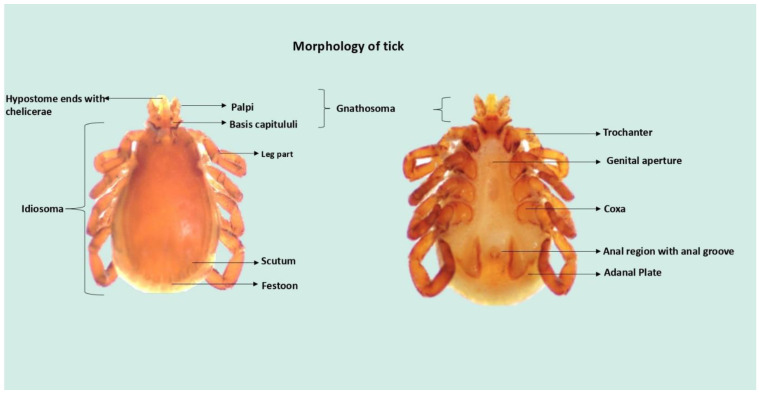
The figure depicts the morphology of the male tick *Rhipicephalus sanguineus*, commonly known as the brown dog tick. In the right panel, a dorsal view highlights the distinct shape of the tick, showcasing a well-defined scutum and festoons. On the left side of the illustration, a ventral view is presented with detailed annotations for the genital aperture, coxa, anal region, and anal groove. Notably, the adanal plate’s shape is emphasized as a key characteristic feature for species identification.

**Figure 3 pathogens-13-00556-f003:**
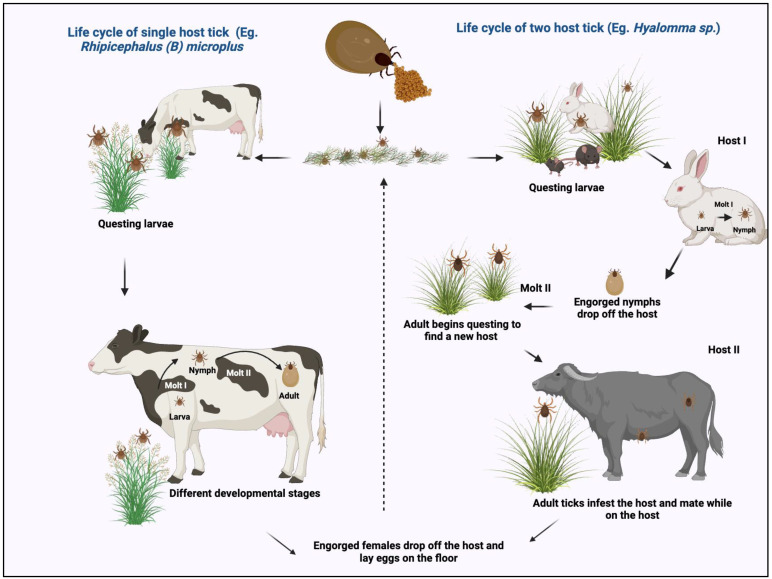
The figure depicts the life cycle of one-host tick (e.g., *Rhipicephalus* (*B*) *microplus*) and two-host tick (*Hyalomma* spp.) [67,94] (illustration created using by BioRender.com, Toronto, Canada).

**Figure 4 pathogens-13-00556-f004:**
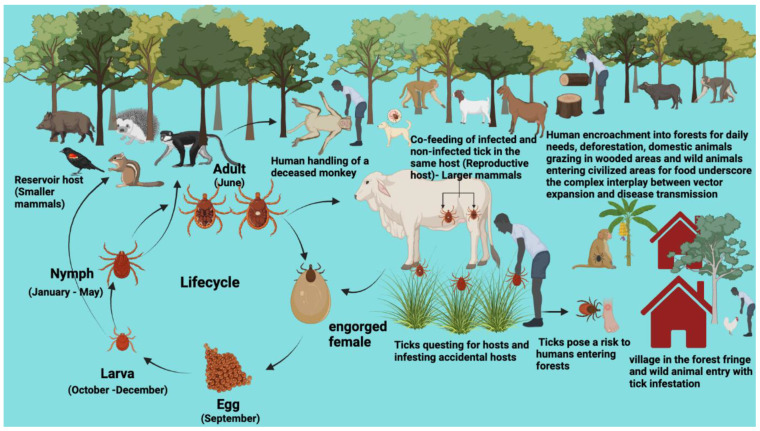
A diagram illustrates the life cycle of the three-host tick *Haemaphysalis spinigera* and its interactions with hosts, highlighting the transmission of the Kyasanur Forest Disease Virus (KFDV). The figure depicts questing ticks and their infestation process involving humans and domestic animals. Special attention is given to the co-feeding phenomenon [14], demonstrating the simultaneous presence of infected and non-infected ticks on the same host (illustration created by using BioRender.com).

**Figure 5 pathogens-13-00556-f005:**
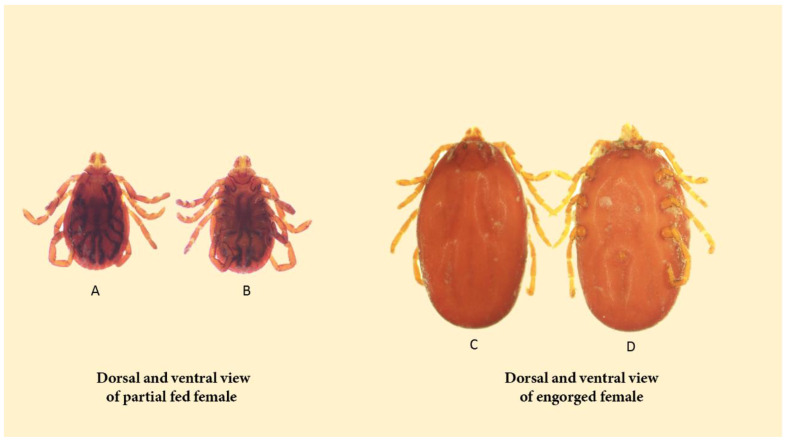
Illustration of dorsal (**A**) and ventral (**B**) view of partially fed and dorsal (**C**) and ventral (**D**) view of engorged female ticks of *Rhipicephalus* spp.

**Figure 6 pathogens-13-00556-f006:**
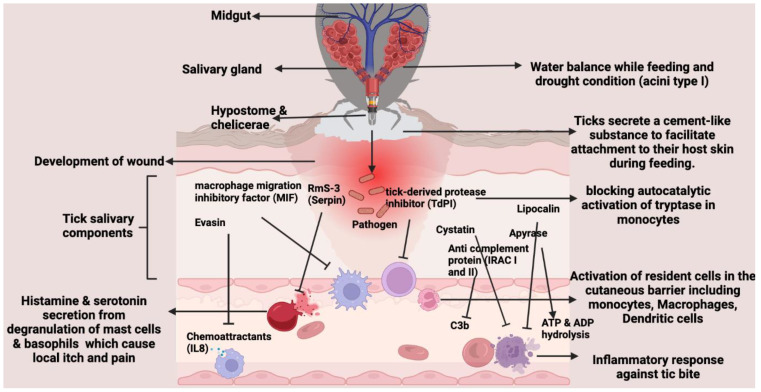
A comprehensive depiction of a tick bite is presented, elucidating the intricate process through which the tick injects various components into its host. The visual narrative highlights the molecular and biochemical aspects of the tick–host interaction, unveiling the complexity inherent in this parasitic relationship. (Illustration created by using BioRender.com).

**Figure 7 pathogens-13-00556-f007:**
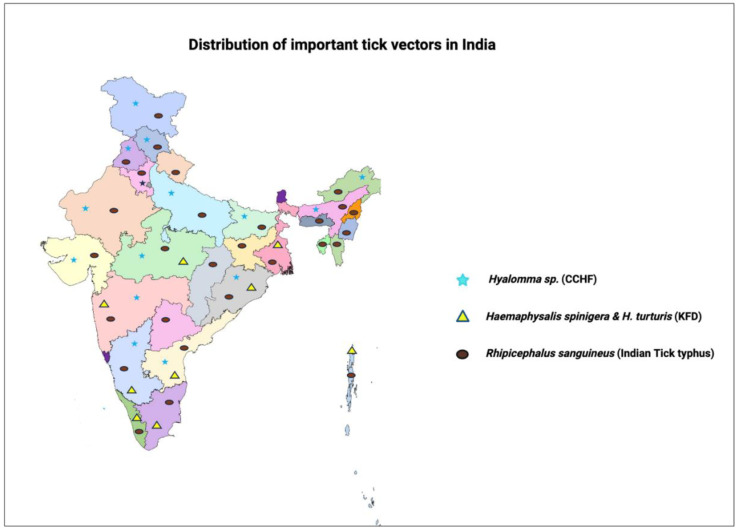
Map of important tick vectors responsible for major TBDs in India.

**Figure 8 pathogens-13-00556-f008:**
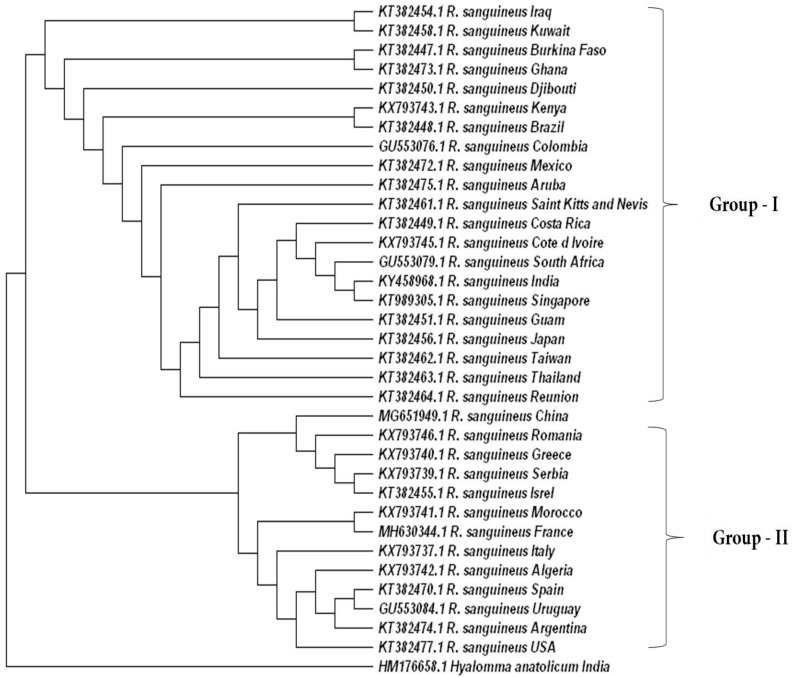
A phylogenetic tree utilizing the Neighbour-Joining method with a Bootstrap value of 1000 was generated from 35 nucleotide sequences of the 16S mitochondrial gene of *Rhipicephalus sanguineus* collected from diverse geographical regions. The analysis excluded any ambiguous positions for each sequence pair through pairwise deletion, resulting in a final dataset of 347 positions. The evolutionary analyses were conducted using MEGA11, and consisted of 347 positions.

**Figure 9 pathogens-13-00556-f009:**
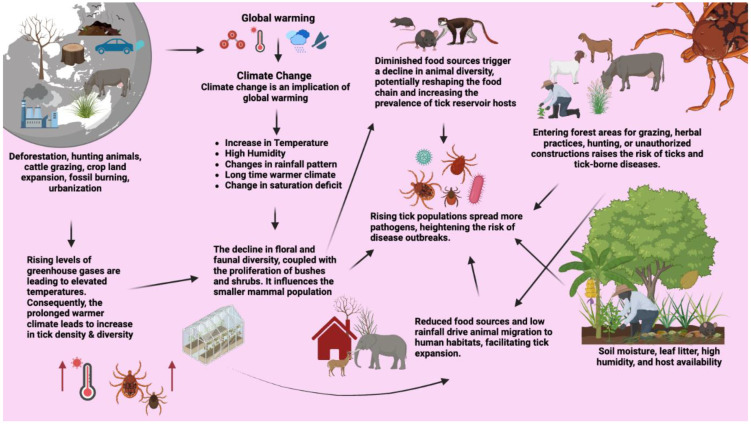
A schematic representation highlights the geographical expansion of ticks, underscoring the significant role played by global warming and climate change in facilitating this spread. It illustrates the impact of climate change on tick habitats and distribution, emphasizing the connection between the loss of floral and faunal diversity and the subsequent increase in tick populations. The figure elucidates the interrelationship between climate change and human activities, emphasizing how anthropogenic factors contribute to the proliferation of ticks. This visual representation offers a comprehensive overview of the intricate dynamics among environmental changes, human activities, and the escalating prevalence of ticks in various regions. (Illustration created by using BioRender.com).

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
