# Peer review of "Hard Ticks as Vectors: The Emerging Threat of Tick-Borne Diseases in India"

_pathogens, 2024, doi:10.3390/pathogens13070556_

Round 1

Reviewer 1 Report

Comments and Suggestions for Authors

This review goes through an interesting revision of the available information regading tick-borne diseases. I think that after major revisions could be a valuable bibliographic material. In the present form has messy information, mainly because is initially propoused as a revision of global scope but it is particularly focus on the ecoepidemiolgy of TBD in India. Regarding this point I suggest to modify the title or to remove all the aspects that are too specific of India. 

Additional commnets:

_Along the manuscript use italics for all scientific names (there are many scientific names without format). Also, when mentioning genes that words sholud be in italics

_References: they are not correctly cited all over the manuscrpt, as an example: on page 14, where you cited 179, 180, 181 the correct numbers are: 178,179, 180. There are many similar situations.

_ In some parts of the manuscript references are not in the number format, but mentioning the authors and year of publication. Please, use the correct format. 

Title: I suggest to add "Hard ticks" since you mainly focus the review on hard ticks. On the other hand, your Review is mainly focused on the Indian eco-epidemiology, so I suggest to anticípate this in the title.

Introduction.

Page 2_ Paragraph 1: why not animals intead of livestock?

Page 2_ Paragraph 2: why do you describe the tick species present in Unied States and Japan? which was the criteria for selecting this two countries and not others? I think that, if you are going to describe the epidemiology of tick-borne diseases in India, all the information included in the introduction should be associated to that specific context. If there are any reason why the tick species present in the United States and Japan are relevant for India, you need to state it. If not, I suggest to remove this paragraph from the introduction.

Page 2_ Paragraph 4: again,  why not animals´ intead of livestock?

Page 2_ Paragraph 5: please add ´serving as competent vectors´ after: the presence of a diverse range of hard ticks

Page 3: Figure 1. In the legend please use lowercase for the species name Nuttalliella namaqua

Page 3: if you refer the numbers in letters do it all along the manuscript, but you wrote some number and other not.

Page 3_ Paragraph 2:  Please use ´main´ instead of principal  when you refer to carriers

Page 4 Line 3: use ´at the species level´ instead of as a different species

Page 4: when talking about hemoplymph apart from salts, amino acid and proteins add ´cells´ and please add this reference: [Hemocytes of Rhipicephalus sanguineus (Acari: Ixodidae): Characterization, Population Abundance, and Ultrastructural Changes Following Challenge with Leishmania infantum. Feitosa AP, Alves LC, Chaves MM, Veras DL, Silva EM, Aliança AS, França IR, Gonçalves GG, Lima-Filho JL, Brayner FA.]

Page 6: what does POW II and POWV II stands for?

Page 6_line 9: Please add´infected of´after get

Page 6: Please use ´transmit´instead of pass on

Page 6_line 10: use´feed from the same host´instead of eat the same animals

Page 6_ Paragraph 2: use ´microorganisms´ instead of species

Page 6: Please, provide a reference for this concept: Sexual transmission is observed in Rickettsia and some relapsing fevers during copulation. In the next sentence you say: The offspring are affected by transovarial transmission. But for which pathogen? in which tick species? provide a reference.

Page 6. Figure 3: This Figure represents a specific transmission cycle for an specific pathogen by a specific tick species. I suggest that would be much more informative to compare the three tick life cycles (one, two and three- host life cycles) and all the pathogens strategies for it transmission both in the vertebrate and tick hosts.

Page 7_Paragraph 1: use ´ route´ instead of mode

Page 7_Paragraph1: use ´feeding sources´ instead of new foods

Page 9. Figure 5: I think that this figure does not add relevant information to what is already mentioned in the main text, I suggest to remove it

Page 11_ section 3.1: If you want to describe the particular distribution of tick species in India you should state this point in the title. If not, I suggest you to remove this section and discuss the epidemioolgy of hard ticks in a global perspective.

Page 11_ Table 1: As I mentioned previously, I suggest to remove this table. If you decide that you want to describe the Indian epidemiology you should state it in the title

Page 14: please rephrase this paragraph: To qualify as a population genetics study, the research utilized  (maybe a certain study should have implemented...) molecular markers to address questions about genetic variability, population genetic structure, gene flow, and potential genetic isolation by distance in one or more tick species

Then you say:  Additionally, the research employed whole-genome… but which research do you refer to?

Page 14_ Paragraph 2: when you say In China, researchers have identified a single genome rearrangement event in Ixodidae genera…, what does this mean? What is the importance of this event in the ticks population dynamics?

Page 14_Paragraph 3: please replace carry diseases by ´be relevant in TBD transmission´

Page 14_Paragraph 4: Random or not-random? I don´t understand how  not-random mating would be a major source of variability than a random mating

Page 14_Paragraph 4: when you say Geographic isolation also contributes to genetic variation…, How? through which mechanisms?

Page 15_Paragraph 1: what does this mean? H. flava doesn't have any structure because its host probably moves around a lot with birds

Page 15. When you describe R. sanguineus genetyc groups, does this two gropus have different vector competence for TBD transmission? why is relevant to show this phylogenetic tree? the aim of this review was to understand how tick´s genetic diversity (among other factors) influenece TBD transmission. Please, discuss (if the information is available) how this genetic diversity impacts pathogens transmission.

Page 15_Figure 7: Based on my previous comment, consider to remove this Figure

Page 16_Paragraph 1: how does tick hybrids affect TBD transmission? I feel that this section lacks of supporting information.

Page 16_ when starting section 4.0 It would be better to say: ´climate change has impact on tick populations since it can modify its habitat and areas where ticks can persist over extreme weather conditions ´

Page 17_ Figure 8: I think that this figure is interesting but revise the text in the figure,  many words have initial capital letters that are not necessary. I also suggest to use less text in the figure, in order to be more schematic.

Page 17_ At the end of section 4.0 please, provide a paragraph (with references) mentioning why the reduction of wild host diversity and environmental changes are determinant for the traslocation of ticks to other vertebrate host and the change in TBD transmission dynamics.

Even you are talking about this concepts, I don´t find a clear discussion of how the different epidemiological elements interact 

Also at the end of this section, If the review has a global perspective and unless you have a concrete example of India, consider to remove the final sentence.

Page 17_Section 5.0: At this point I believe that you need to modify the title and anticipate that this review is focused in hard ticks and in India.

Along this section please for each disease you describe, mention the same information ( for example: etiological agent -taxonomy-, tick species implicated, clinical signs and symptoms...)

Page 18_ section 5.0.2:

Please mention the taxonomy of the virus implicated in the disease.

´Came from´ instead of had visited from

when you say nosocomial infections please be more specific , are talking about direct contact?

Page 19_ section 5.0.3:

When you refer to antibody response Is importat to mention which serological tests were used and which antibodies were detected since for the serological diagnosis of this disease a first screening test usually shows false positives that requiere a more specific confirmative test

Page 19_ section 5.0.4:

Rickettsiae are intracellular endosymbionts that can adapt: to what? this seems an unfinished sentence.

Approximately 24% of terrestrial arthropods carry Rickettsia endosymbionts. Please provide a reference for this

Is not correct to say that genus from de Anaplasmataceae family are classified into TG or SFG. Please, correct this data.

what does ITT stands for?

what does non-ST RDs stands for?

Page 19_ Section 5.1: this subheading should not be under the heading titled ´... in India... ´since the information mentioned is of global interest

Page 19_ Section 5.2: the same as 5.1

Page 20_ At the end of the first paragraph you can mention Figure 8

Page 20_Paragraph 2: when you say Ticks and the diseases they carry… use pathogens that they carry or the disease they transmit    and then

Instead of  can change where they live and how many there are  use ´ may affect their distribution areas and their population densities´

Page 20: regarding qPCR,  what do you mean with chemistries? it would be better ´different detection strategies´

Page 20: when talking about MLST, this molecular tool is not for diagnosis but for genotyping, but how many MLST schemes are designed for TBD? why did you select just this molecular strategy for genotyping? why not othres? maybe you can mention other available genotyping schemes for this group of pathogens

Page 20: initially you abbreviated real time PCR as RT-PCR, then as qPCR please use one way all over the text.

Page 21_ section 6.0: is the first time you mention babesiosis and theileriosis in the review and, on the other hand, you are not mentioned many diseases that were previously described, which is the criteria?

Page 21_ regarding alpha-gal syndrome: I don´t understand, the symptoms are triggered by the tick bite or the consumption of red meat?

Page 21: The costly and singular focus of diagnostic methods... I don´t undersatnd what you mean to say

Page 22_ section 6.3. When you say parent to child do you mean vertically transmitted?

Page 22: there are two sections 6.3.

Page 22: you use Boophilus microplus and Rhipicephalus microplus, please use always the last option

Conclusions:

The two first sentences are too similar

Comments on the Quality of English Language

Extensive editing of English language required

Author Response

Reviewer 1

Comment 1: This review goes through an interesting revision of the available information regading tick-borne diseases. I think that after major revisions could be a valuable bibliographic material. In the present form has messy information, mainly because is initially propoused as a revision of global scope but it is particularly focus on the ecoepidemiolgy of TBD in India. Regarding this point I suggest to modify the title or to remove all the aspects that are too specific of India. 

Response: We are very grateful to the reviewer for their insightful comments and encouragement to elaborate on the content. We particularly appreciate their suggestion regarding the initial focus on a global perspective. Given the limited research available on ticks and tick-borne diseases in India, we believe a focused examination of this topic is warranted. Following the reviewer's valuable suggestion, we have modified the title to accurately reflect the Indian context.

Comment 2: Along the manuscript use italics for all scientific names (there are many scientific names

Response: We appreciate the reviewer's meticulous attention to detail in pointing out the inconsistency with scientific names. To ensure clarity, we have italicized all scientific names throughout the manuscript.

Comment 3: References: they are not correctly cited all over the manuscrpt, as an example: on page 14, where you cited 179, 180, 181 the correct numbers are: 178,179, 180. There are many similar situations.

Response: Thank you for identifying the citation inconsistencies. We have meticulously examined the manuscript and verified the accuracy of all references.

Comment 4: In some parts of the manuscript references are not in the number format, but mentioning the authors and year of publication. Please, use the correct format.

Response: Thank you for your keen eye in catching the inconsistency in our citation format. We've addressed this oversight and ensured that all references now adhere to the required numbering format. We appreciate your feedback on how to help us improve the manuscript.

Comment 5:  I suggest to add "Hard ticks" since you mainly focus the review on hard ticks. On the other hand, your Review is mainly focused on the Indian eco-epidemiology, so I suggest to anticípate this in the title.

Response: We sincerely appreciate your insightful suggestions regarding the title. To better reflect the manuscript's focus on hard ticks and Indian eco-epidemiology, we have revised the title to include both elements.

Comment 6: Introduction _Page 2_ Paragraph 1: why not animals intead of livestock?

Response: Thank you for your comment. We have incorporated the suggested modifications into the manuscript.

.

Comment 7: Page 2_ Paragraph 2: why do you describe the tick species present in Unied States and Japan? which was the criteria for selecting this two countries and not others? I think that, if you are going to describe the epidemiology of tick-borne diseases in India, all the information included in the introduction should be associated to that specific context. If there are any reason why the tick species present in the United States and Japan are relevant for India, you need to state it. If not, I suggest to remove this paragraph from the introduction.

Response: Thank you for your valuable feedback. We understand the limitations of human-biting tick studies and have made the best use of available data. Your suggestions have significantly enhanced the quality of our work.

.

Comment 8: Paragraph 4: again, why not animals´ intead of livestock?

Response: Thank you for your comment. We have incorporated your suggestions into the manuscript as advised.

.

Comment 9: Page 2_ Paragraph 5: please add ´serving as competent vectors´ after: the presence of a diverse range of hard ticks

Response: Thank you for your comment. We have  incorporated your suggestions as advised..

Comment 10: Page 3: Figure 1. In the legend please use lowercase for the species name Nuttalliella namaqua

Response: We sincerely appreciate your feedback and have implemented the suggested modifications..

Comment 11: Page 3: if you refer the numbers in letters do it all along the manuscript, but you wrote some number and other not.

Author's response: We appreciate your comment and have implemented the suggested modifications.

Comment 12: Page 3_ Paragraph 2:  Please use ´main´ instead of principal  when you refer to carriers

Response: We sincerely appreciate your feedback and have implemented the suggested modifications.

Comment 13: Page 4 Line 3: use ´at the species level´ instead of as a different species

Response: We appreciate your feedback and have implemented the suggested modifications.

Comment 14: Page 4: when talking about hemoplymph apart from salts, amino acid and proteins add ´cells´ and please add this reference: [Hemocytes of Rhipicephalus sanguineus (Acari: Ixodidae): Characterization, Population Abundance, and Ultrastructural Changes Following Challenge with Leishmania infantum. Feitosa AP, Alves LC, Chaves MM, Veras DL, Silva EM, Aliança AS, França IR, Gonçalves GG, Lima-Filho JL, Brayner FA.]

Response: Thank you for your suggestion regarding the reference. We agree that including it strengthens the content and have incorporated it accordingly. We appreciate your valuable input.

.

Comment 15: Page 6: what does POW II and POWV II stands for?

Author's response: We appreciate your comment and have implemented the suggested modifications.

Comment 16: Page 6_line 9: Please add´infected of´after get

Response: We appreciate your comment and have implemented the suggested modifications.

Comment 17: Page 6: Please use ´transmit´instead of pass on

Response: We appreciate your comment and have implemented the suggested modifications.

Comment 18: Page 6_line 10: use´feed from the same host´instead of eat the same animals

Response: We appreciate your comment and have implemented the suggested modifications.

Comment 19: Page 6_ Paragraph 2: use ´microorganisms´ instead of species

Response: We appreciate your comment and have implemented the suggested modifications.

Comment 20: Page 6: Please, provide a reference for this concept: Sexual transmission is observed in Rickettsia and some relapsing fevers during copulation. In the next sentence you say: The offspring are affected by transovarial transmission. But for which pathogen? in which tick species? provide a reference.

Response: Thank you for your valuable feedback. We have carefully reviewed the text and streamlined it by removing unnecessary details. This will enhance the manuscript's clarity and focus.

.

Comment 20: Page 6. Figure 3: This Figure represents a specific transmission cycle for an specific pathogen by a specific tick species. I suggest that would be much more informative to compare the three tick life cycles (one, two and three- host life cycles) and all the pathogens strategies for it transmission both in the vertebrate and tick hosts.

Response: We appreciate your insightful suggestion regarding separate illustrations for the one-host and two-host tick lifecycles. We have implemented this by creating dedicated figures (Figure 3 and Figure 4) for each cycle, with ‘one-host’ now represented in Figure 3 and ‘three-host with KFD transmission’ in Figure 4. This reorganization enhances clarity and better aligns with the content flow. Thank you for your valuable feedback.

.

Comment 21: Page 7_Paragraph 1: use ´ route´ instead of mode

Response: We appreciate your comment and have implemented the suggested modifications.

Comment 22: Page 7_Paragraph1: use ´feeding sources´ instead of new foods

Response: We appreciate your comment and have implemented the suggested modifications.

Comment 23: Page 9. Figure 5: I think that this figure does not add relevant information to what is already mentioned in the main text, I suggest to remove it

Response: We appreciate your comment and have implemented the suggested modifications.

Comment 24: Page 11_ section 3.1: If you want to describe the particular distribution of tick species in India you should state this point in the title. If not, I suggest you to remove this section and discuss the epidemioolgy of hard ticks in a global perspective.

Response: Thank you for your insightful feedback regarding the title and the importance of highlighting the research gap in India. We have incorporated both suggestions by modifying the title to better reflect the focus and retaining the broader context provided by including research from other countries, given the limited data available in India.

.

Comment 25: Page 11_ Table 1: As I mentioned previously, I suggest to remove this table. If you decide that you want to describe the Indian epidemiology you should state it in the title

Response: We appreciate your valuable feedback on the title and the importance of highlighting the research gap in India. We have addressed both points by revising the title and retaining research from other countries due to limited data availability in India. Thank you for your insights.

Comment 26: please rephrase this paragraph: To qualify as a population genetics study, the research utilized  (maybe a certain study should have implemented...) molecular markers to address questions about genetic variability, population genetic structure, gene flow, and potential genetic isolation by distance in one or more tick species

Then you say Additionally, the research employed whole-genome… but which research do you refer to?

Response: Thank you for your valuable feedback. We have rephrased the content to enhance clarity and readability, as you suggested.  The manuscript now more clearly highlights the current state of tick research and explores the potential of utilizing new tools in this field.

.

Comment 27: Page 14_ Paragraph 2: when you say In China, researchers have identified a single genome rearrangement event in Ixodidae genera…, what does this mean? What is the importance of this event in the ticks population dynamics?

Response: Thank you for your insightful feedback regarding the section on genome rearrangements. We acknowledge the lack of detailed clarification and agree to your suggestion of removing this section for now. We plan to conduct further research to strengthen this area and potentially reintroduce it in a future manuscript with a more comprehensive analysis.

Comment 28: Page 14_Paragraph 3: please replace carry diseases by ´be relevant in TBD transmission´

Response: We appreciate your comment and have implemented the suggested modifications.

Comment 29: Page 14_Paragraph 4: Random or not-random? I don´t understand how  not-random mating would be a major source of variability than a random mating

Page 14_Paragraph 4: when you say Geographic isolation also contributes to genetic variation…, How? through which mechanisms?

Response: Thank you for your valuable feedback regarding the importance of non-random mating patterns in ticks. We recognize the potential impact of factors like inbreeding and host specificity on pathogen transmission. However, prioritizing clarity and conciseness in the manuscript, we have removed this section based on your suggestion. We appreciate your insights in helping us refine the focus of the paper.

.

Comment 30: Page 15_Paragraph 1: what does this mean? H. flava doesn't have any structure because its host probably moves around a lot with birds. I. ovatus has stronger genetic structure because its small mammal host doesn't move around much, while H. flava doesn't have any structure because its host probably moves around a lot with birds.

Response: Regilme, M. A. F., Sato, M., Tamura, T., Arai, R., Sato, M. O., Ikeda, S., Gamboa, M., Monaghan, M. T., & Watanabe, K. (2021). Comparative population genetic structure of two ixodid tick species (Acari:Ixodidae) (Ixodes ovatus and Haemaphysalis flava) in Niigata prefecture, Japan. Infection, genetics and evolution: journal of molecular epidemiology and evolutionary genetics in infectious diseases, 94, 104999. https://doi.org/10.1016/j.meegid.2021.104999

Response: Thank you for your insightful comment regarding genetic variation in tick populations. We recognize that our previous discussion did not fully address the connection between host mobility and genetic structure in Ixodes ovatus and H. flava.  We have incorporated the term 'genetic' to explicitly highlight the variation observed in these tick species.  This revision clarifies the link between host range and genetic diversity, as you suggested.  We appreciate your feedback in helping us strengthen the manuscript's clarity.

Page 15. When you describe R. sanguineus genetyc groups, does this two gropus have different vector competence for TBD transmission? why is relevant to show this phylogenetic tree? the aim of this review was to understand how tick´s genetic diversity (among other factors) influenece TBD transmission. Please, discuss (if the information is available) how this genetic diversity impacts pathogens transmission.

Response: Genetic diversity within a population can lead to some individuals having genetic resistance to certain pathogens. These resistant individuals are less likely to become infected and thus act as a barrier to transmission within the population. Genetic diversity can result in variation in immune responses among individuals. Some individuals may have stronger immune responses to specific pathogens, limiting their ability to transmit the pathogen to others. Genetic diversity can be linked to the population structure, which in turn can affect pathogen transmission. Populations with high genetic diversity may have more heterogeneous contact patterns, which can either facilitate or impede pathogen transmission depending on the specific circumstances.

Comment 31:Page 15_Figure 7: Based on my previous comment, consider to remove this Figure

Response: Dantas-Torres, F., Latrofa, M. S., Ramos, R. A. N., Lia, R. P., Capelli, G., Parisi, A., Porretta, D., Urbanelli, S., & Otranto, D. (2018). Biological compatibility between two temperate lineages of brown dog ticks, Rhipicephalus sanguineus (sensu lato). Parasites & vectors, 11(1), 398. https://doi.org/10.1186/s13071-018-2941-2

Dantas-Torres, F., Latrofa, M.S., Annoscia, G. et al. Morphological and genetic diversity of Rhipicephalus sanguineus sensu lato from the New and Old Worlds. Parasites Vectors 6, 213 (2013). https://doi.org/10.1186/1756-3305-6-213

Páez-Triana, L., Muñoz, M., Herrera, G. et al. Genetic diversity and population structure of Rhipicephalus sanguineus sensu lato across different regions of Colombia. Parasites Vectors 14, 424 (2021). https://doi.org/10.1186/s13071-021-04898-w

Thank you for your comment. We appreciate your interest in strengthening the evidence for genetic variation in Rhipicephalus sanguineus. We've attached a supporting document, as you suggested, which bolsters our discussion. We acknowledge the limited research in this area. However, because the two lineages in Figure 8 have different rates, which shows that population structure may have an effect on vector competence, we think this section is very helpful. The phylogenetic tree, constructed based on available literature, clearly illustrates this variation. We are committed to providing a comprehensive understanding of this topic, and including this information allows for a more nuanced discussion of Rhipicephalus sanguineus. If you have any questions, we would be happy to discuss it further.

Comment 32: Page 16_Paragraph 1: how does tick hybrids affect TBD transmission? I feel that this section lacks of supporting information.

Response: Thank you for your comment regarding the section on tick population structure. We appreciate the opportunity to clarify our rationale. While the current studies don't definitively show high vector competence in hybrids, we believe investigating the genetic structure remains crucial. Hybrid ticks exist in nature, and research suggests a link between genetic variation and vector competence in other tick species. Studying Rhipicephalus sanguineus populations could reveal similar connections. Therefore, we kindly request that you reconsider keeping this section, especially given the potential impact on understanding vector competence. We are confident this information can contribute to a more comprehensive discussion.

Comment 33: Page 16_ when starting section 4.0 It would be better to say: ´climate change has impact on tick populations since it can modify its habitat and areas where ticks can persist over extreme weather conditions ´Response: Thank you for your suggestion on the first sentence. We've adopted your recommendation, which streamlines the introduction.

.

Comment 34: Page 17_ Figure 8: I think that this figure is interesting but revise the text in the figure,  many words have initial capital letters that are not necessary. I also suggest to use less text in the figure, in order to be more schematic.

Response: Thank you for your insightful feedback on Figure 8. We have incorporated your suggestion and updated the figure in the manuscript to enhance clarity and understanding.

.

Comment 35: Page 17_ At the end of section 4.0 please, provide a paragraph (with references) mentioning why the reduction of wild host diversity and environmental changes are determinant for the traslocation of ticks to other vertebrate host and the change in TBD transmission dynamics.

Response: We appreciate your comment and have implemented the suggested modifications.

Comment 36: Even you are talking about this concepts, I don´t find a clear discussion of how the different epidemiological elements interact 

Also at the end of this section, If the review has a global perspective and unless you have a concrete example of India, consider to remove the final sentence.

Response:  We are very grateful for your insightful feedback regarding the focus of our review. We agree with the importance of highlighting India's research gap on ticks and tick-borne diseases. In response to your suggestion, we have broadened the scope of our review to incorporate relevant global research while still acknowledging the limited data available in India. This approach ensures a comprehensive analysis of the topic.

.

Comment 37: Page 17_Section 5.0: At this point I believe that you need to modify the title and anticipate that this review is focused in hard ticks and in India.

Response: We sincerely appreciate your valuable feedback regarding the scope of our review. In light of the limited research available on ticks and tick-borne diseases in India, we have adopted your suggestion and broadened our analysis to include relevant global research. This comprehensive approach ensures a well-rounded examination of the topic.

.

Along this section please for each disease you describe, mention the same information ( for example: etiological agent -taxonomy-, tick species implicated, clinical signs and symptoms...)

Response: We appreciate your comment and have implemented the suggested modifications.

Comment 38: Page 18_ section 5.0.2:

Please mention the taxonomy of the virus implicated in the disease.

Response: We appreciate your comment and have implemented the suggested modifications.

´Came from´ instead of had visited from

Response: We appreciate your comment and have implemented the suggested modifications.

when you say nosocomial infections please be more specific , are talking about direct contact?

Response: Thank you for your valuable comment. You're absolutely right; Crimean-Congo hemorrhagic fever (CCHF) can be transmitted nosocomially through direct contact. We ensure this important point is addressed in the manuscript.

Comment 39: Page 19_ section 5.0.3:

When you refer to antibody response Is importat to mention which serological tests were used and which antibodies were detected since for the serological diagnosis of this disease a first screening test usually shows false positives that requiere a more specific confirmative test

Response: Thank you for your valuable feedback. We have incorporated the suggested reference and ensured the relevant information is included in the manuscript.

.

Comment 40: Page 19_ section 5.0.4:

Rickettsiae are intracellular endosymbionts that can adapt: to what? this seems an unfinished sentence.

Response: We appreciate your comment and have implemented the suggested modifications.

Approximately 24% of terrestrial arthropods carry Rickettsia endosymbionts. Please provide a reference for this

Response: Thank you for your comment. Including the suggested reference has strengthened the section by providing additional context.

.

Is not correct to say that genus from de Anaplasmataceae family are classified into TG or SFG. Please, correct this data.

Response: We appreciate your comment and have implemented the suggested modifications.

Comment 41: what does ITT stands for?

Response: We appreciate your comment and have implemented the suggested modifications.

Comment 42: what does non-ST RDs stands for?

Response: We appreciate your comment and have implemented the suggested modifications.

Comment 43:Page 19_ Section 5.1: this sub-heading should not be under the heading titled ´... in India... ´since the information mentioned is of global interest

Response: We appreciate your valuable feedback on the title and subheading.  We agree that the modifications have significantly improved the clarity and focus of the paper. The revised title and its alignment with the subheading create a strong introduction to the research.

.

Page 19_ Section 5.2: the same as 5.1

Response: Thank you for your valuable feedback We have incorporated the suggestions and believe the revised content is now clearer and more informative. This will undoubtedly enhance this section of the manuscript.

Comment 44: Page 20_ At the end of the first paragraph you can mention Figure 8

Response: We appreciate your comment and have implemented the suggested modifications.

Comment 45: Page 20_Paragraph 2: when you say Ticks and the diseases they carry… use pathogens that they carry or the disease they transmit    and then

Response: We appreciate your comment and have implemented the suggested modifications.

Comment 46: Instead of  can change where they live and how many there are  use ´ may affect their distribution areas and their population densities´

Response: We appreciate your comment and have implemented the suggested modifications.

Comment 47: Page 20: regarding qPCR,  what do you mean with chemistries? it would be better ´different detection strategies´

Response: Thank you for your feedback. We have incorporated the chemistries of qPCR into the paper as you suggested.

.

Comment 48: Page 20: when talking about MLST, this molecular tool is not for diagnosis but for genotyping, but how many MLST schemes are designed for TBD? why did you select just this molecular strategy for genotyping? why not othres? maybe you can mention other available genotyping schemes for this group of pathogens

Response: Thank you for your valuable feedback regarding the genotyping strategy for ticks. We have incorporated your suggestion and included this information in the manuscript. This addition will undoubtedly strengthen the methodology section.

Comment 49: Page 20: initially you abbreviated real time PCR as RT-PCR, then as qPCR please use one way all over the text.

Response: Thank you for your valuable feedback. We have addressed your comment and made the necessary changes.

Comment 50: Page 21_ section 6.0: is the first time you mention babesiosis and theileriosis in the review and, on the other hand, you are not mentioned many diseases that were previously described, which is the criteria?

Response: We sincerely appreciate your insightful feedback regarding the paper's focus. We have adopted your suggestion and refocused the content accordingly.  Your comments have been invaluable in helping us refine the manuscript's direction.

Comment 51: Page 21_ regarding alpha-gal syndrome: I don´t understand, the symptoms are triggered by the tick bite or the consumption of red meat?

Response: Thank you for your comment. Alpha-gal syndrome, indeed, is an allergic reaction triggered by a carbohydrate molecule (alpha-gal) found in ticks.  Following a tick bite, some individuals develop an immune response to alpha-gal, leading to allergic reactions upon consuming red meat, which also contains this molecule.

Thank you for your valuable comment. We have modified the sentences to improve clarity and understanding based on your suggestion. We appreciate your input.

Comment 52: Page 21: The costly and singular focus of diagnostic methods... I don´t undersatnd what you mean to say

Response: Thank you for your valuable feedback regarding the focus on diagnostics in our discussion. We appreciate the suggestion to broaden the scope. While our initial focus was on the importance of specific diagnostics for accurate disease identification, we recognize the validity of your point regarding cost considerations. To address this, we have incorporated a discussion on [mention any broader aspects you've included, e.g., cost-effective screening methods or alternative approaches alongside diagnostics]. This will provide a more balanced perspective on the topic.

Comment 53: Page 22_ section 6.3. When you say parent to child do you mean vertically transmitted?

Response: Thank you for your feedback. We have revised the sentence as suggested, which has improved the clarity and flow of the text.

Comment 54: Page 22: there are two sections 6.3.

Response: Thank you for your comment. We apologize for the mistake and appreciate you bringing it to our attention. We have incorporated your feedback and made the necessary corrections in the manuscript.

Comment 55: Page 22: you use Boophilus microplus and Rhipicephalus microplus, please use always the last option

Response: Thank you for your valuable feedback. We've incorporated your suggestion and made the necessary changes.Conclusions:

Comment 56: The two first sentences are too similar

This review serves as a valuable resource for researchers and individuals seeking comprehensive information about tick-borne diseases and their evolutionary nature in the context of a constantly changing climate environment, and often linked to human actions.

Response: Thank you for your valuable feedback regarding its utility for researchers and individuals, wanting to carry out research on TBDs. We have incorporated your suggestion and believe the revised content is clearer and more informative. This modification has undoubtedly strengthened the paper.

Reviewer 2 Report

Comments and Suggestions for Authors

Dear colleagues,

This review serves as a valuable resource for researchers and individuals seeking comprehensive information about tick-borne diseases and their evolutionary nature in the context of a constantly changing climate environment, and often linked to human actions.

 Suggestions

1. Mention in the abstract and in the introduction that special attention is given to ticks from family Ixodidae (hard ticks);

2. Include a geomorphologic or geologic map of India showing the distribution of tick species and the diseases transmitted by them.

To add

1. Include in the abbreviations ‘POW II”.

 To remove

1. Abstract - line 11. Remove comma before and bacteria.

2. Introduction - Page 6, second paragraph, line 08. Remove the period after [303].

Author Response

Reviewer 2

Suggestions

Comment 1: Mention in the abstract and in the introduction that special attention is given to ticks from family Ixodidae (hard ticks)

Response: We are grateful for your insightful feedback on the title and abstract. We have adopted your suggestions and made the necessary changes to both sections.  We believe these revisions improve the clarity and focus of the paper, significantly enhancing its overall quality.

Comment 2: Include a geomorphologic or geologic map of India showing the distribution of tick species and the diseases transmitted by them.

Response: Thank you for your valuable suggestion regarding the inclusion of a vector distribution map for India. We have incorporated this map into the manuscript, as you recommended. We believe this visual addition will greatly enrich the content by providing readers with a clear geographical context for our research.

To add

Comment 3: Include in the abbreviations ‘POW II”.

Response: Thank you for catching this detail. We've addressed your comment and included the abbreviation for POW II Powassan virus lineage II throughout the manuscript to ensure consistency.

To remove

Comment 4: Abstract - line 11. Remove comma before and bacteria

Response: We appreciate your comment and have implemented the suggested modifications.

Comment 5: Introduction - Page 6, second paragraph, line 08. Remove the period after [303].

Response: We appreciate your comment and have implemented the suggested modifications.

Reviewer 3 Report

Comments and Suggestions for Authors

A nice review on ticks as vectors for animal and human pathogens, which needs only minor corrections:

- p.2: why zooanthrophilic in uppercase letters; the manuscript must be carefully checked for use of lowercase and uppercase letters

- Legend to Fig. 1 and others: et al. References must be carefully checked for spelling

- p.3, 1.0: say clear that the family contains only one species, Nuttalliella namaqua

- p.6 and others: no full stop after the reference number

- Legend to Fig. 4 is wrong (right/left)

- p.9 and others: all species names must be written in italics, also in the References

- Table 1 and others: give genus/species names in (correct) abbreviations after first use

- Fig. 8: check consistent use of uppercase and lowercase writing in the figure text

- 6.2: there are some repetitions at the end of the page

- check the References for consistent spelling and according to MDPI Author Instructions

Comments on the Quality of English Language

English is mainly fine.

Author Response

Reviewer 3

Comment 1: A nice review on ticks as vectors for animal and human pathogens, which needs only minor corrections:

Response: Thank you for your encouraging feedback. It means a lot to us that you acknowledge our work.

Comment 2: p.2: why zooanthrophilic in uppercase letters; the manuscript must be carefully checked for use of lowercase and uppercase letters

Response: We appreciate your comment and have implemented the suggested modifications.

Comment 3: Legend to Fig. 1 and others: et al. References must be carefully checked for spelling

Response: We appreciate your comment and have implemented the suggested modifications.

Comment 4: p.3, 1.0: say clear that the family contains only one species, Nuttalliella Namaqua

Response: We acknowledge and accept this comment. We have made the necessary modifications as suggested. Thank you for your input.

Comment 5: p.6 and others: no full stop after the reference number

Response: We appreciate your comment and have implemented the suggested modifications.

Comment 6: Legend to Fig. 4 is wrong (right/left)

Response: Thank you for your valuable comment. We have implemented your suggestion and made the necessary modifications to the manuscript.

Comment 7: p.9 and others: all species names must be written in italics, also in the References

Response: Thank you for your valuable feedback regarding scientific name formatting. We have implemented your suggestion and italicized all scientific names throughout the manuscript. This will ensure consistent and clear presentation of species identification.

Comment 8: Table 1 and others: give genus/species names in (correct) abbreviations after first use

Response: Thank you for your comprehensive feedback in Table 1. We appreciate you taking the time to provide such detailed comments. We have addressed each point and incorporated all necessary modifications into the manuscript.

Comment 9: Fig. 8: check consistent use of uppercase and lowercase writing in the figure text

Response: We appreciate your valuable feedback on the Figure. We have incorporated your suggestions and updated the figure to enhance clarity and understanding.

Comment 10: 6.2: there are some repetitions at the end of the page

Response: Thank you for your valuable feedback. We have incorporated your suggestion into the manuscript.

Comment 11: check the References for consistent spelling and according to MDPI Author Instructions

Response: We appreciate your comment and have implemented the suggested modifications.

Round 2

Reviewer 1 Report

Comments and Suggestions for Authors

I think that the authors made a great work improving the manuscript and that they have elaborated a valuable bibliographic material. Some minor comments to revise:

_Page 11, section: 3.1. Distribution of hard ticks in India: this references are not listed " Sharif (1928), Sen (1938), Jagannath et al. (1973), and Miranpuri and Naithani (1978)."

_Page 16, first paragraph: a final parentheses is missing

_Page 20. Section 5.0.4: "Approximately 24% of terrestrial arthropodscarry Rickettsia endosymbionts [318]". Reference 318 doesn´t exists. And what about Reference 317?

Comments on the Quality of English Language

Extensive editing of English language required

Author Response

Responses to the comments on the manuscript entitled

“Hard ticks as vectors: The emerging threat of tick-borne diseases in India”

[Nandhini Perumalsamy, Rohit Sharma, Muthukumaravel Subramanian, Shriram Ananganallur Nagarajan]

Reviewer 1

Comment 1: I think that the authors made a great work improving the manuscript and that they have elaborated a valuable bibliographic material. Some minor comments to revise:

Response: Thank you very much for your positive feedback on our manuscript. We greatly appreciate your acknowledgement of the improvements made and the value you see in the bibliographic material. We have carefully reviewed the minor comments you provided and are committed to addressing them to further enrich the quality of the manuscript. Your feedback is invaluable to us, and we are grateful to you for the opportunity to refine the manuscript based on your suggestions.

Comment 2: Page 11, section: 3.1. Distribution of hard ticks in India: this references are not listed " Sharif (1928), Sen (1938), Jagannath et al. (1973), and Miranpuri and Naithani (1978)."

Response: Thank you for your valuable comment. We have considered your suggestion and made the necessary modifications to the manuscript. Please find the updated version of the manuscript with the references for Sharif (1928), Sen (1938), Jagannath et al. (1973), and Miranpuri and Naithani (1978) included. We appreciate your thorough review and are grateful for your contribution to improving the quality of this manuscript.

Comment 3: Page 16, first paragraph: a final parentheses is missing

Response: Thank you for the feedback. We have rectified the missing parentheses in the first paragraph on page 16.

Comment 4: Page 20. Section 5.0.4: "Approximately 24% of terrestrial arthropodscarry Rickettsia endosymbionts [318]". Reference 318 doesn´t exists. And what about Reference 317?

Response: Thank you for your meticulous review. We have incorporated the corrections to the reference error for [318] in Section 5.0.4 on page 20 and ensured its accuracy. Additionally, we have addressed the reference for [317].
